# Combining sophisticated fast FLIM, confocal microscopy, and STED nanoscopy for live-cell imaging of tunneling nanotubes

Magalie Bénard[1] , Christophe Chamot[1], Damien Schapman[1], Aurélien Debonne[1,2], Alexis Lebon[1], Fatéméh Dubois[3,4], Guénaëlle Levallet[3,4] , Hitoshi Komuro[1], Ludovic Galas[1]

**Cell-to-cell communication via tunneling nanotubes (TNTs) is a challenging topic with a growing interest. In this work, we proposed several innovative tools that use red/near-infrared dye labeling and employ lifetime-based imaging strategies to investigate the dynamics of TNTs in a living mesothelial H28 cell line that exhibits spontaneously TNT1 and TNT2 subtypes. Thanks to a fluorescence lifetime imaging microscopy module being integrated into confocal microscopy and stimulated emission depletion nanoscopy, we applied lifetime imaging, lifetime dye unmixing, and lifetime denoising techniques to perform multiplexing experiments and time-lapses of tens of minutes, revealing therefore structural and functional characteristics of living TNTs that were preserved from light exposure. In these conditions, vesicle-like structures, and tubular- and round-shaped mitochondria were identified within living TNT1. In addition, mitochondrial dynamic studies revealed linear and stepwise mitochondrial migrations, bidirectional movements, transient backtracking, and fission events in TNT1. Transfer of Nile Red–positive puncta via both TNT1 and TNT2 was also detected between living H28 cells.**

## Introduction

The direct connection between distant cells via tunneling nanotubes (TNTs) is now well documented in many cell lines (Pinto et al, 2020; Roehlecke & Schmidt, 2020). The architecture of TNTs and the identification of their components were usually studied on fixed samples through electron and advanced light microscopy approaches (Lou et al, 2017; Dubois et al, 2020; Cordero Cervantes & Zurzolo, 2021). Scanning electron microscopy (SEM), transmission electron microscopy (TEM), and associated variations for 3D imaging including focused ion beam SEM (FIB-SEM), serial-sectioning TEM (ssTEM), and cryo-electron tomography (cryo-ET) have revealed the structural features of TNTs in various cultured cells (Cordero Cervantes & Zurzolo, 2021). Through SEM and ssTEM, the characteristics of single TNTs (diameter, length, endpoints) were determined in neuronal cell lines including PC12 (Rustom et al, 2004) and STHdh (Sharma & Subramamiam, 2019) cells, and in epithelial cell lines, for example, A549 (Kumar et al, 2017), MDCK (Kumar et al, 2017), HBEC-3, and H28 (Dubois et al, 2018, 2020) cells. In addition, the elegant combining of electron and fluorescence microscopies, for example, cryo-correlative wide-field microscopy/TEM or confocal microscopy/FIB-SEM, was used to identify individual TNTs and vesicles within bundles or bunches of multiple TNTs in neuronal and stromal cell lines (Kolba et al, 2019; Sartori-Rupp et al, 2019). The structural analyses of TNTs were also performed through super-resolution approaches including stimulated emission depletion (STED) nanoscopy and structured illumination microscopy (SIM). Through STED nanoscopy, two types of TNTs, named TNT1 and TNT2, were discovered in PC12 (Bénard et al, 2015) and HBEC-3 (Dubois et al, 2018) cells. TNT1 was thicker and longer than TNT2. Containing both actin and tubulin, TNT1 interconnected cell bodies through large trumpet-shaped origins. TNT2 only contained actin and linked cell bodies through tiny branched attachments (Bénard et al, 2015; Dubois et al, 2018). Extra-centrosomes were also detected within TNT1 that connect HBEC-3 cells (Dubois et al, 2021). In cancer urothelial cells, 3D-SIM experiments revealed the presence of microtubules (MTs) and intermediate filaments (IFs) within TNTs, with MTs potentially forming a helical wrapping around IFs (Resnik et al, 2018). SIM imaging also highlighted the role of $Ca^{2+}$/calmodulin-dependent protein kinase II (CaMKII) in the regulation of TNT formation in CAD cells (Vargas et al, 2019).

To fully understand how TNTs are formed, as well as the mechanisms of transfer between cells (Sharma & Subramamiam, 2019), a detailed study of TNTs in living cells is necessary. However, live-cell imaging poses technical challenges because of its

---

[1]University Rouen Normandie, INSERM, CNRS, Normandie Université, HeRacLeS US51, UAR2026, PRIMACEN, Rouen, France   [2]University Rouen Normandie, INSERM, Normandie Université, UMR1245, Rouen, France   [3]Université de Caen Normandie, CNRS, Normandie Université, ISTCT UMR6030, Caen, France   [4]Service d'Anatomie et Cytologie Pathologiques, CHU de Caen, Caen, France

Correspondence: ludovic.galas@univ-rouen.fr; magalie.benard@univ-rouen.fr

thinness, photosensitivity, and fragility, making it difficult to employ potentially damaging methods (Dubois et al, 2020). The TNT dynamics have been widely investigated through classical bright-field and fluorescence microscopies. As recently revealed through wide-field microscopy, centrosomes or mitochondria were transferred via TNTs between bronchial epithelial cell lines (Dubois et al, 2018) or between glioblastoma cells and surrounding non-tumor astrocytes (Valdebenito et al, 2021), respectively. In lung cancer cells, the regulatory action of a splicing isoform of the neuronal guidance gene *MICAL2* (MICAL2PV) on the formation of TNTs and on the mitochondrial trafficking was demonstrated through confocal microscopy analysis (Wang et al, 2021; Dong et al, 2023). By increasing temporal resolution through spinning disk confocal microscopy, the crucial role of lysosomes in α-synuclein spreading via TNTs was identified in CAD-HeLa co-culture (Dilsizoglu Senol et al, 2021). Interestingly, fluorescence resonance energy transfer (FRET) approaches were also developed to study the TNT biogenesis and the transfer mechanisms. FRET-based Rho GTPase biosensors revealed a distinct activation pattern of Cdc42 and Rac1 at the base and within TNTs in a macrophage cell line (Hanna et al, 2017). In addition, two-photon excitation fluorescence lifetime imaging microscopy (FLIM)-FRET was used to detect material transport via TNTs in ovarian cancer cells (Wang et al, 2021). However, live-cell imaging of TNTs through STED nanoscopy has been limited to a few studies mainly because of the deleterious impact of high-energy depletion laser (Bénard et al, 2015; Reindl et al, 2019). In particular, the application of a 592-nm STED depletion laser to Alexa Fluor 488–conjugated WGA-labeled PC12 cells induced membrane distortion and TNT disruption during time-lapse experiments (Bénard et al, 2015).

The aim of this work was to develop advanced light microscopy and nanoscopy approaches to precisely study TNTs in living cells. In this context, living H28 cells, which spontaneously expressed type 1 and type 2 TNTs, were labeled with red/near-infrared fluorescent probes to monitor membrane, cytoskeleton component, and mitochondrial and lipid-rich organelle dynamics. By taking advantages of the instrumental performances and considering the sample preservation via limitation of the light exposure (Bénard et al, 2021), we conducted time-lapse imaging using fluorescence lifetime-based strategies, including FLIM-associated confocal and FLIM-associated STED imaging.

# Results

## Mesothelial H28 cells as a model to study TNT characteristics between cancer cells

As previously described (Dubois et al, 2018), cultured H28 cells spontaneously generated a high proportion of TNT subtypes, namely, TNT1 and TNT2, making them a good model to study cell-to-cell communication. Therefore, we started our study by conducting triple-labeling through immunocytochemical experiments and STED nanoscopy imaging to reveal the cytoskeleton components in fixed H28 cells. Using two depletion laser wavelengths (592 and 775 nm), this super-resolution

approach allowed the fine localization of actin, tubulin, and vimentin within TNT subtypes. Because a 592-nm depletion laser resulted in rapid dye photobleaching (~23.5% after nine frames, *$P < 0.05$ versus ctrl) compared with a 775-nm depletion laser, which had no significant effect on fluorescent dye intensity (Fig S1A), sequential acquisitions were performed. Respectively revealed by antibodies coupled to Alexa Fluor 594 and Atto 647N, vimentin and tubulin were first detected through STED nanoscopy thanks to a 775-nm depletion laser. STED imaging of F-actin filaments was then obtained with Alexa Fluor 488–conjugated phalloidin depleted via a 592-nm wavelength. TNT1 were relatively long processes (40–60 µm) containing actin, tubulin, and vimentin (Fig 1A), and they connected cells without any substratum contact (Fig S1B). Connecting close H28 cells, short TNT2 (<15 µm) only contained actin and were devoid of tubulin and vimentin (Fig 1B). Besides the structural characterization of TNTs in fixed cells mostly using antibodies and toxins, the next challenge was to have access to high- and super-resolution imaging in order to study TNTs in living cells while reducing the phototoxicity.

## Association of red/near-infrared dye labeling with lifetime-based imaging strategies as innovative tools to characterize TNTs between living cells

As described previously (Bénard et al, 2021), labeling cellular organelles and membranes with red/near-infrared dyes, along with appropriate instrumental key elements including a water immersion objective with high transmission, an integrated FLIM module and sensitive hybrid detectors have enabled flexible and fast imaging of living cells with a low irradiance and a reduced cell phototoxicity. Because of offering multiple options in confocal and STED acquisition modes, the integrated FLIM module was definitively a central and revolutionary tool in our strategy and Fig 2 is proposed as a guide for readers to perform labeling and long-lasting time-lapse experiments for studying structural and functional aspects of living TNTs including organelle trafficking. Collected in one spectral window, the informative lifetime data may provide mono- or multi-component(s) for a single dye via a "lifetime imaging mode." In order to reduce significantly the light exposure during multi-labeling experiments, several spectrally close fluorochromes could be illuminated with a single excitation wavelength, detected in a unique emission band, and separated according to their distant lifetime values via a "lifetime dye unmixing mode." Furthermore, in order to improve the quality of STED images, an integrated FLIM module may also elicit a high signal-to-noise ratio via a "lifetime denoising mode." In our study, "lifetime denoising" included (i) removal of uncorrelated STED process photons (background subtraction, weighting of lifetime) and (ii) signal smoothing through a wavelet filter. The lifetime imaging, lifetime dye unmixing, and lifetime denoising modes could therefore be relevant to investigate the dynamics of TNTs with one or several red/near-infrared dyes including possibly STED-compatible dyes (Fig 2). The advantages and limitations of the different modes described in Fig 2 will be considered in the Discussion section.

**A**

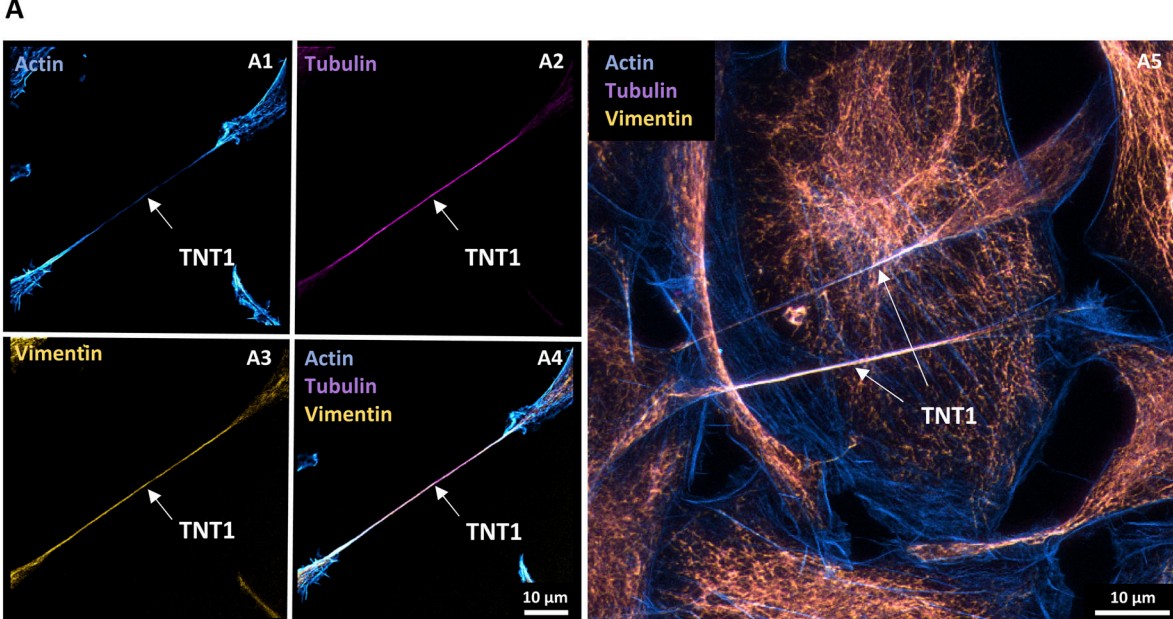

**STED nanoscopy - 3 dyes - Fixed H28 cells**

**B**

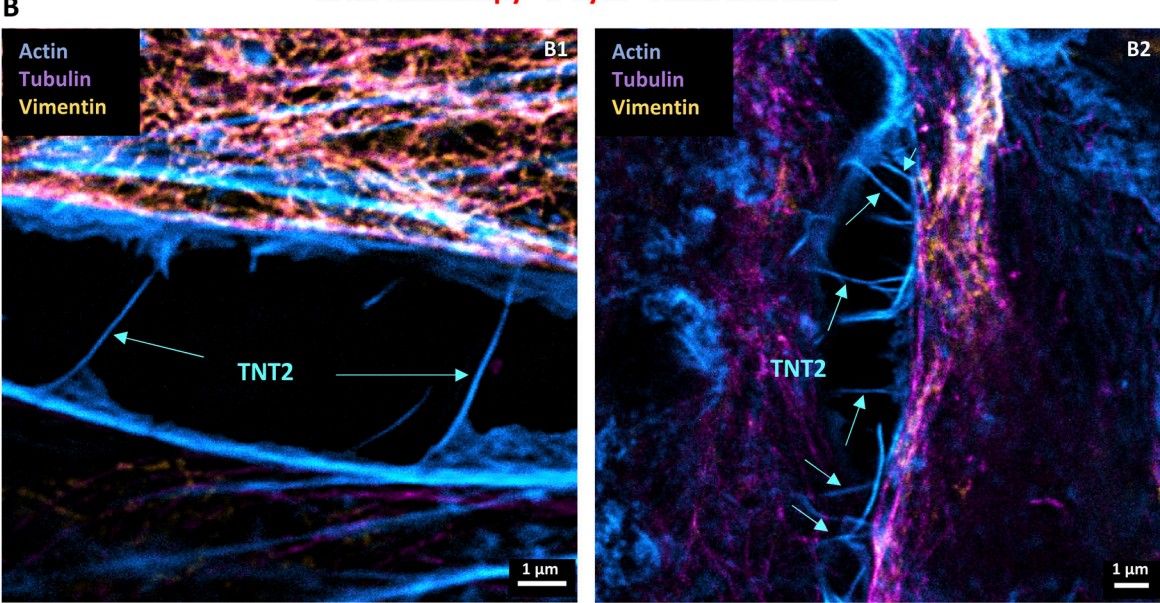

**Figure 1. Localization of cytoskeleton proteins in fixed tunneling nanotube (TNT)-connected H28 cells through stimulated emission depletion (STED) nanoscopy and lifetime denoising.**

Imaging of triple-stained H28 cells was performed sequentially through lifetime denoising and STED nanoscopy. First, Alexa Fluor 594–conjugated vimentin (ex 600 nm, 7% WLL laser power—HyD-S 595–630 nm, yellow LUT) and Atto 647N–conjugated tubulin (ex 647 nm, 7% WLL laser power—HyD-X 660–710 nm, magenta LUT) were sequentially revealed through STED nanoscopy thanks to depletion with 20% of a 775-nm pulsed laser. Secondly, Alexa Fluor 488–conjugated phalloidin signal (ex 488 nm, 5% WLL laser power—HyD-S, cyan LUT) was depleted with 40% of a 592-nm continuous laser to detect F-actin filaments through STED nanoscopy. **(A)** Sequential scans and maximum intensity projection revealing actin (A1, **blue**), tubulin (A2, **magenta**), and vimentin (A3, **yellow**) in TNT1 (white arrow) and resulting overlay image (A4). Sequential stack-by-stack (15 z; 0.14 μm/step) scans in 3D STED (30%) and resulting overlay image showing actin, tubulin, and vimentin in TNT1 (white arrows) after maximum intensity projection (A5). **(B)** Overlay images showing the distribution of actin in TNT2 (blue arrows) and the absence of tubulin and vimentin proteins.

### FLIM-integrated confocal microscopy revealed TNT dynamics in living H28 cells labeled with 1, 2, or 3 red/infrared dyes

As a first step, we conducted experiments by activating a single excitation wavelength and a single detector in order to reduce the image acquisition time and the exposure of TNTs to light.

Using single-dye labeling such as SPY650-FastAct for actin or Alexa Fluor 633–conjugated WGA for membranes, both TNT1 and TNT2 were revealed in living H28 cells through FLIM-integrated confocal microscopy (Fig 3A). With a length of several tens of micrometers, TNT1 were relatively long and single (Fig 3 A1 and A2), whereas TNT2 were shorter and multiple (Fig 3 A3). Because SPY650-

**Figure 2. Central role of a fluorescence lifetime imaging microscopy–integrated module in live-cell imaging of a tunneling nanotube through confocal microscopy and stimulated emission depletion nanoscopy.**
Sophisticated methods to image living tunneling nanotubes. Strategies for functional considerations, multiplexing, and signal-to-noise ratio increase were presented with respect to post-processing modes of lifetime data including "lifetime imaging," "lifetime dye unmixing," and "lifetime denoising."

## Sophisticated methods to image living TNTs

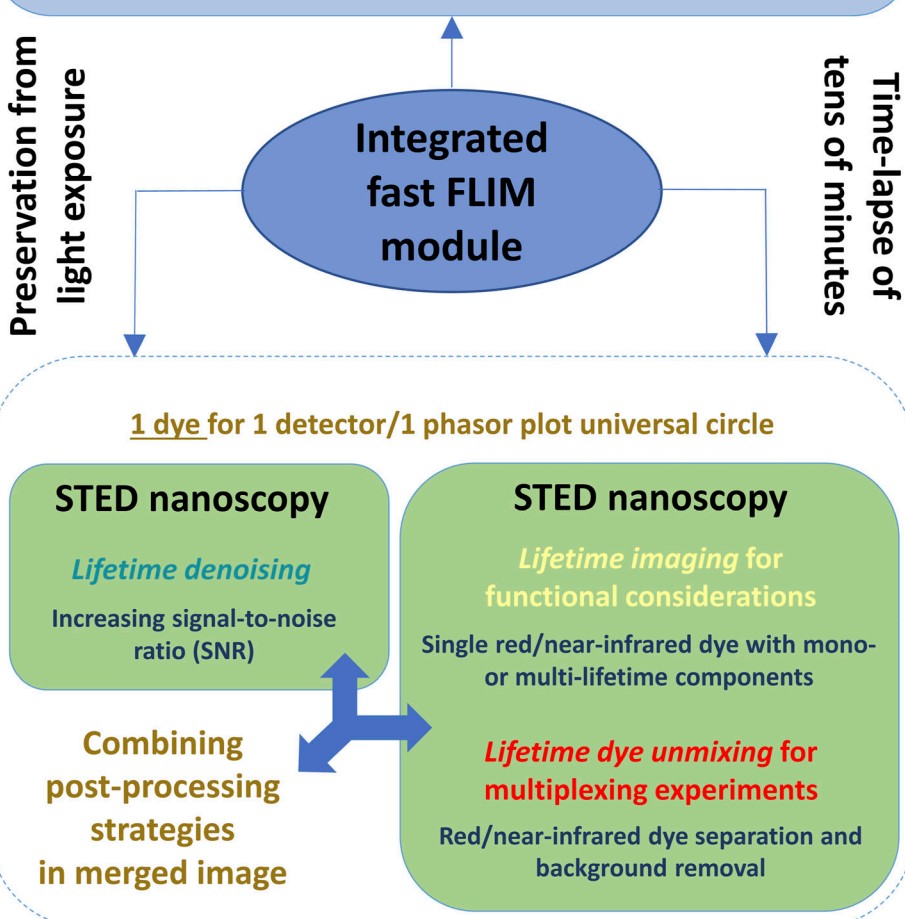

FastAct labeling displayed a mono-component of lifetime, the resulting lifetime image was homogeneous (Figs 3 A1 and S2A). In contrast, Alexa Fluor 633–conjugated WGA lifetimes were distributed within a rainbow look-up table, revealing the diversity of intracellular environment associated with cellular membrane compartments including vesicles (Figs 3 A2 and S2B). Interestingly, the dispersion of Alexa Fluor 633–conjugated WGA lifetimes was broader along TNT1 and within the large bud-shaped protrusion (Fig 3 A2) compared with that observed in TNT2 (Fig 3 A3). TNT1 and TNT2 lifetime phasor plot profiles were different (Fig S2B and C).

Indeed, TNT1 ROI displayed higher Alexa Fluor 633–conjugated WGA lifetime values compared with TNT2 ROI (Fig S2 B3 and C3). Altogether, these data suggest the presence of WGA-positive vesicle-like structures in TNT1 but not in TNT2.

When double-labeling of H28 cells was performed with two spectrally close dyes, for example, SPY650-FastAct for actin and LBL-Dye M715 for mitochondria, optimization of the configuration for fast FLIM-integrated confocal microscopy and simultaneous acquisition was required. The activation of a single HyD-X detector and a single excitation wavelength of 650 nm with a very low laser

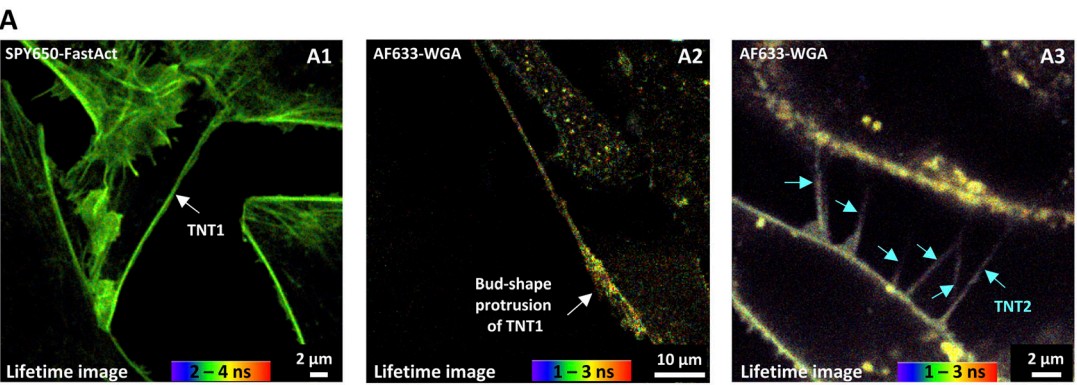

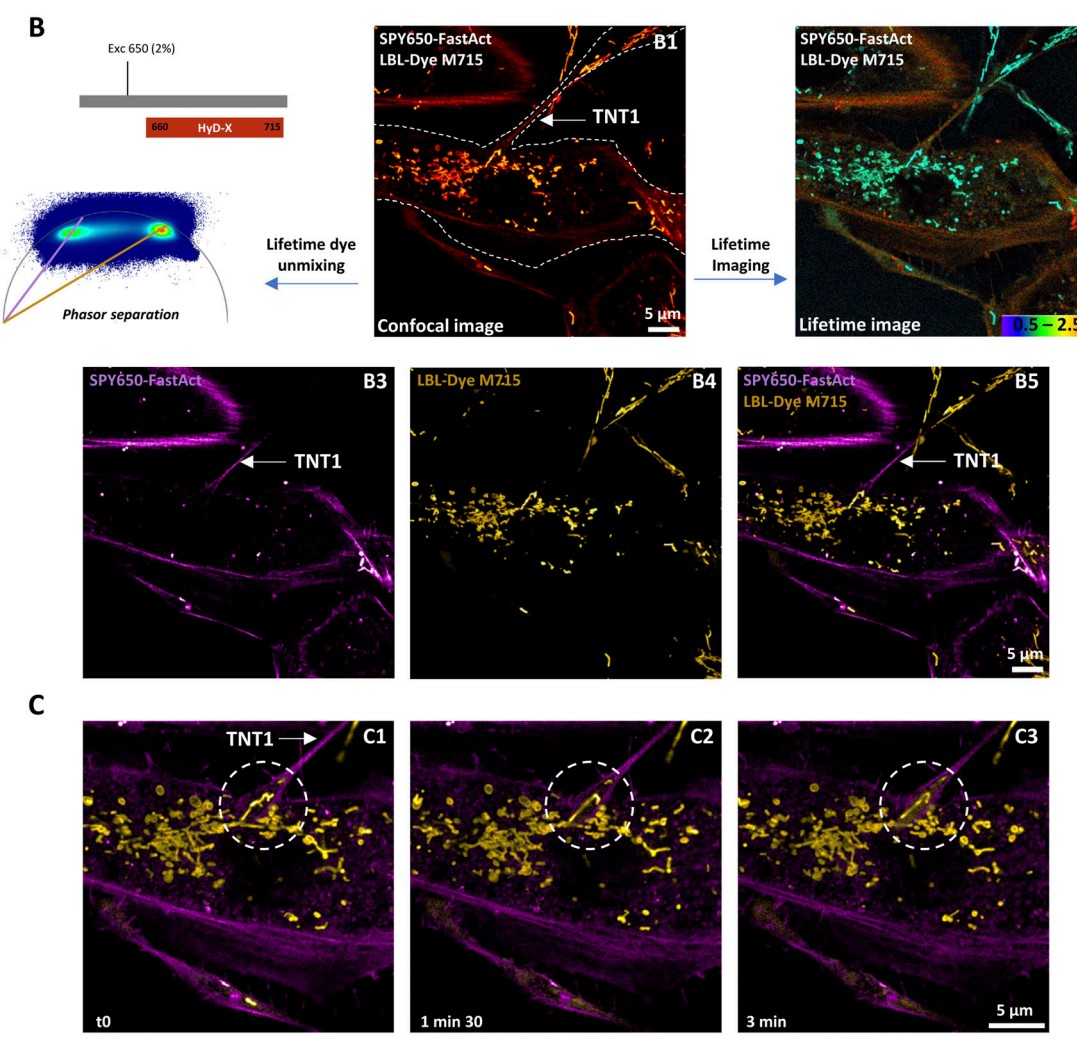

**Figure 3. Live-cell imaging of tunneling nanotubes (TNTs) between H28 cells through red and near-infrared dye labeling and fluorescence lifetime imaging microscopy–integrated confocal microscopy.**

**(A)** Combining confocal and lifetime microscopies for single-labeling experiments. SPY650-FastAct (actin, ex 650 nm, 2% WLL laser power—HyD-X 660–715 nm) (A1) or Alexa Fluor 633 (AF633)–conjugated WGA (membrane, ex 638 nm, 5% WLL laser power—HyD-X 640–670 nm) with maximum intensity projection of a five-image z-stack, 0.68 μm/step (A2), and maximum intensity projection of a two-image z-stack, 0.27 μm/step (A3), stain revealed TNT1 (white arrows) and TNT2 (blue arrows). **(B)** Simultaneous imaging of H28 cells labeled with SPY650-FastAct and LBL-Dye M715 (mitochondria) (ex 650 nm, 2% WLL laser power—HyD-X 660–715 nm) through confocal microscopy (B1), lifetime imaging (B2), or lifetime dye unmixing after phasor plot separation of actin filaments (B3) and mitochondrial signals (B4), overlay of lifetime dye unmixing images (B5). **(C)** Regions of interest in overlays (white dotted circles) showing the dynamics of actin cytoskeleton and mitochondria during a 3-min time-lapse (C1–C3). A white dotted line delimitated the cell body and associated TNT1.

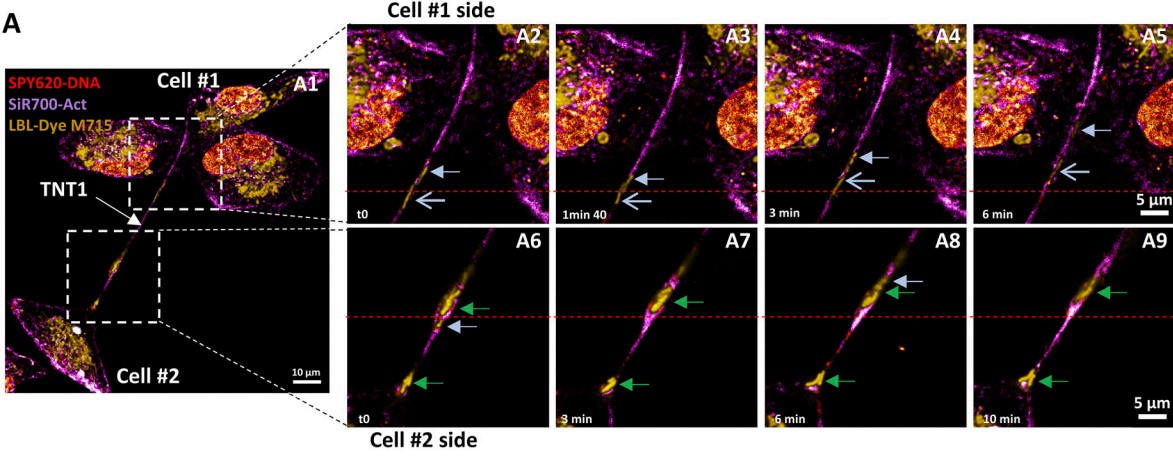

## FLIM-integrated confocal microscopy - 3 dyes - living H28 cells

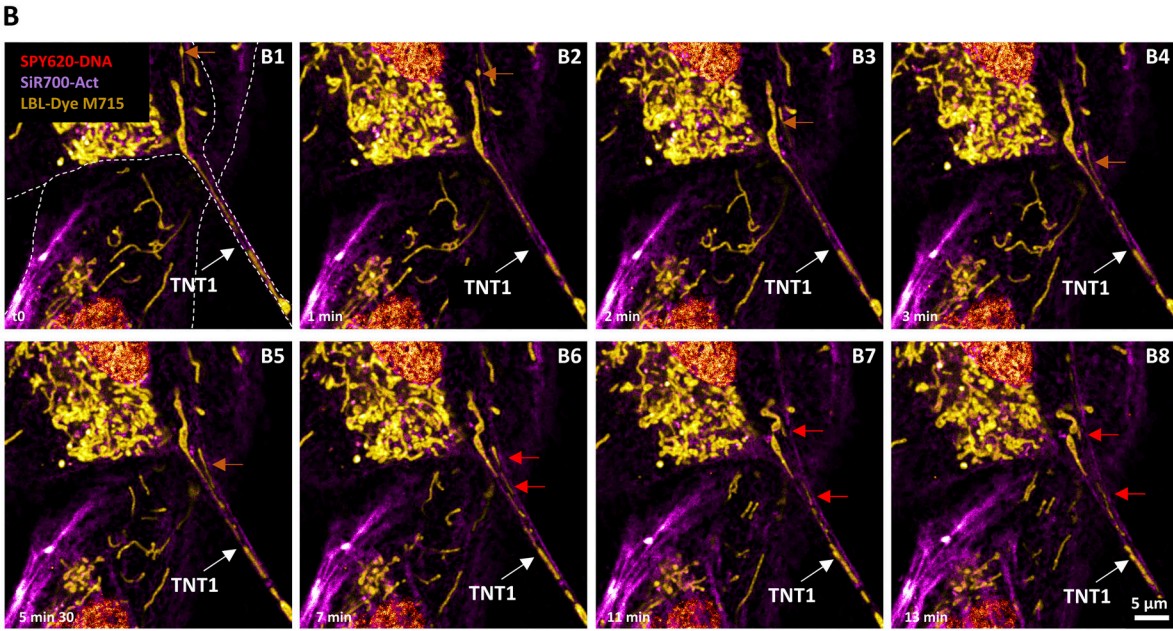

**Figure 4.  Live-cell imaging of TNT1 between triple-labeled H28 cells through fluorescence lifetime imaging microscopy–integrated confocal microscopy and lifetime dye unmixing.**

**(A, B)** Simultaneous imaging in a single detector (ex 618 nm, 5% WLL laser power—HyD-X 625–715 nm) of three dyes including SPY620-DNA (red), SiR700-Act (magenta), and LBL-Dye M715 (yellow) that labeled TNT1 and H28 cell bodies. Dyes were separated via lifetime unmixing. **(A)** Time-lapse acquisition (10 min, 1 image/5 s) in regions of interest close to Cell #1 (A2–A5) and Cell #2 (A6–A9) bodies. Short light blue and green arrows indicated mitochondrial migration and membrane bulging/mitochondrial reorientation, respectively. **(B)** Mitochondrial dynamics (20 min time-lapse—Video 3, 1 image/30 s, orange arrows) including fission and bilateral migration (red arrows) within TNT1 and cell body. Reference red dotted lines facilitated the observation of positions during migration.

power (2%) led to a minimum irradiance and to "lifetime dye unmixing" after "lifetime imaging" (Fig 3B). In lifetime imaging (rainbow LUT from 0.5 to 2.5 ns), mitochondria were detected with short lifetime values (cyan-green) of LBL-Dye M715, whereas actin cytoskeleton was revealed by higher lifetime values of SPY650-FastAct within the orange-red range. To improve dye separation, the application of the phasor plot tool allowed the identification of two spots of lifetime values at 1.2 and 2.8 ns for LBL-Dye M715 and SPY650-FastAct, respectively (Figs S3A and 3 B3 and B4). In particular, an overlay image obtained from "lifetime dye unmixing" showed the presence and movements of mitochondria within

trumpet-shaped TNT1 originating from the H28 cell body (Fig 3C, Video 1).

When triple-labeling of the H28 cell was performed with spectrally close dyes, such as SPY620-DNA for the nucleus, SiR700-Act for actin, and LBL-Dye M715 for mitochondria, a similar strategy was followed. Because the lifetime of SPY620-DNA ($\tau$ = 2.56 ns) was too close to the SPY650-FastAct lifetime ($\tau$ = 2.8 ns), SiR700-Act ($\tau$ = 1.644 ns) was used to facilitate the lifetime unmixing (Fig S3B). In this case, the activation of a single HyD-X detector and a single excitation wavelength of 618 nm with a low laser power (8%) led again to a minimum irradiance and to "lifetime dye unmixing" after

"lifetime imaging" (Fig S3C). The DNA labeling was very helpful to identify cell bodies (Cell #1 and Cell #2) that were connected by TNT1 (Fig 4A). Under these conditions, the dynamics of mitochondria within TNT1 were observed (Fig 4) through time-lapse experiments lasting over 10 min (1 image every 5 s; Video 2) and with minimal photobleaching (Fig S4A). Mitochondria migrated with different rate profiles from 0.8 to 8.14 µm/min (Fig 4A, blue arrows). Both TNT1 and mitochondrion shapes slowly changed, including plasma membrane bulging and mitochondrial reorientation (Fig 4 A6–A9, green arrows). Interestingly, a mitochondrion fission could also be detected, leading to bidirectional movements along TNT1 (red arrows indicated separation, Fig 4B, Video 3). The median speed of mitochondria within TNT1 was 1.435 µm/min (min = 0.8 µm/min; max = 8.14 µm/min; n = 24), whereas the first and third quartile values were 0.935 and 4.525, respectively.

Collectively, these data indicated that TNT1 imaging through FLIM-integrated confocal microscopy could be performed for more than 10 min when living H28 cells were triple-labeled.

### FLIM-integrated STED nanoscopy and lifetime denoising revealed TNT dynamics in living H28 cells labeled with 1 or 2 red/infrared dyes

Because of the depletion laser power limitations, STED nanoscopy for living samples was a matter of debate; however, recent instrumental developments, including the 775-nm depletion laser wavelength, the enhanced detector sensitivity, the lifetime data, and the post-processing software, offered new applications (Bénard et al, 2021). As a second step, we developed strategies to image TNTs through time-lapse experiments using STED nanoscopy. In order to preserve the viability of the sample during long time-lapse imaging, a low 775-nm depletion laser power was combined with background subtraction, lifetime weighting, and an additional wavelet filter–based lifetime denoising approach that improved the signal-to-noise ratio. When applying only 2% of a 775-nm depletion laser power, LBL-Dye M715 fluorescence intensity was unaffected during the first 90 sample scans and only showed a non-significant 10% reduction after 230 sample scans (Fig S4B).

Using 775-nm STED-compatible Nile Red, which is usually described as a vital intracellular lipid marker, both TNT1 and TNT2 were revealed and time-lapses were performed over several minutes (Fig 5). Nile Red–positive signals were detected along the length of TNTs with wavelet/accordion-like appearances in TNT1 and straight aspects in TNT2 (Fig 5). Nile Red also labeled branched attachments of TNT2 that strengthened TNT2 to the cell body (green arrows, Fig 5B). In the cytoplasm of cell bodies, Nile Red also labeled punctiform organelles that were sparsely found within TNT1 (Fig 5A) and TNT2 (Fig 5B). Time-lapse imaging clearly illustrated the movements of Nile Red–labeled puncta within TNT1 (red circles, 0.53 µm/min) and TNT2 (red circles, 4.6 µm/min) toward H28 cell bodies (Video 4 and Video 5, respectively). The average speed of Nile Red–positive puncta in TNT1 and TNT2 was 0.76 ± 0.24 µm/min (n = 5) and 4.86 ± 0.6 µm/min (n = 5), respectively.

Using 775-nm STED-compatible LBL-Dye M715, the mitochondrial tracking within TNT1 was performed, leading to the identification of different rate profiles depending on the organelle shape and location (Fig 6 A). At the TNT1 basis, a tubular-shaped mitochondrion

was first immobile before outlining a rapid round trip (5.9 µm/min) (Fig 6 A2). Along TNT1 but near the Cell #1 body, both round- and tubular-shaped mitochondria were observed (Fig 6 A1). Round-shaped mitochondria migrated either linearly (0.85 µm/min) (Fig 6 A3) or in a stepwise manner (maximum of 3.4 µm/min) (Fig 6 A4) toward the Cell #2 body (Video 6). Rare transient backtracking events could also be detected. Close to the TNT1 endpoint (Fig 6B), a tubular-shaped mitochondrion displayed linear and moderately rapid movement (2.8 µm/min) before reaching the Cell #2 body.

Taken together, these data indicated that TNT imaging through STED nanoscopy could be performed during 20-min time-lapses when H28 cells were mono-labeled.

We also developed strategies to perform STED imaging of TNTs between double-labeled cells. Because lifetime data contained in images generated through FLIM-integrated STED nanoscopy were post-processed, the acquisition time and consequently the light exposure were reduced during time-lapses. In addition, lifetime denoising, lifetime imaging, and lifetime dye unmixing could be properly used or combined to improve the signal-to-noise ratio, to identify the different lifetimes, or to separate the dye signals in order to reveal a TNT in living H28 cells (Figs 2 and S5). Because a STED depletion laser altered the fluorescence lifetime characteristics (Gonzalez Pisfil et al, 2022) (Fig S5A), the achievement of multi-labeling experiments with spectrally close dyes became more difficult and only a strong enhancement may reveal specific signals, for example, actin (Fig S5B). Therefore, a dedicated "HyD detector–phasor plot circle" for each dye was necessary to perform accurate post-processing.

When double-labeling of H28 cells was performed with two spectrally close dyes, such as SPY650-FastAct for actin and LBL-Dye M715 for mitochondria, optimization of the FLIM-integrated STED nanoscopy configuration and simultaneous acquisition with two detectors was required. The activation of a HyD-S detector and an excitation wavelength of 645 nm with 6% laser power was dedicated to SPY650-FastAct, whereas the activation of a HyD-X detector and an excitation wavelength of 698 nm with 12% laser power was dedicated to LBL-Dye M715 (Fig S6). A single 775-nm STED depletion laser (2%) was used. The SPY650-FastAct signal in HyD-S was denoised (STED lifetime denoising; Fig S6 A1) and used for the final merged image, whereas the LBL-Dye M715 signal in HyD-X was spectrally contaminated with the SPY650-FastAct signal (Fig S6 A2–A5). Consequently, the LBL-Dye M715 lifetime signal detected in HyD-X was "decontaminated" via lifetime unmixing and used for the final merged image (Fig S6 A9).

Suggesting different metabolic states of mitochondria, the different MitoTracker Red (MTR) lifetime values were revealed through FLIM-integrated STED nanoscopy both in H28 cell bodies and within TNT1 (Fig 7A). In particular, MTR-labeled mitochondria with short (1.3–1.8 ns, blue arrows) and high (3.6–3.7 ns, green arrows) lifetimes were found all along TNT1 connecting H28 living cells. When double-labeling of H28 cells was performed with MTR and SiR700-Act, optimization of the configuration for FLIM-integrated STED nanoscopy and simultaneous acquisition with two detectors was required (Fig 7B). The activation of a HyD-S detector and an excitation wavelength of 590 nm with 1% laser power was dedicated to MTR, whereas the activation of a HyD-X detector and an excitation wavelength of 698 nm with 6% laser power was dedicated to SiR700-

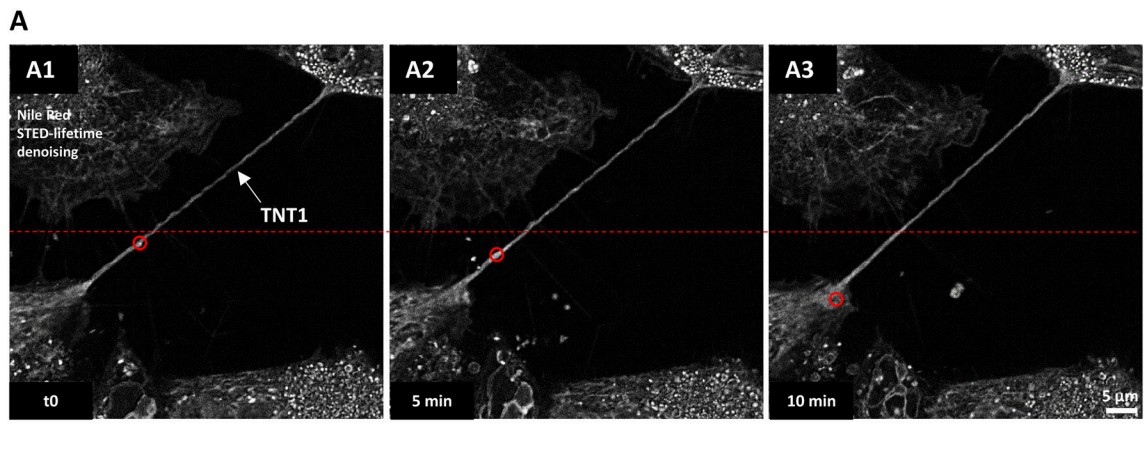

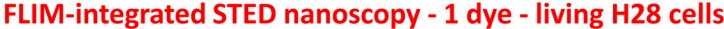

**FLIM-integrated STED nanoscopy - 1 dye - living H28 cells**

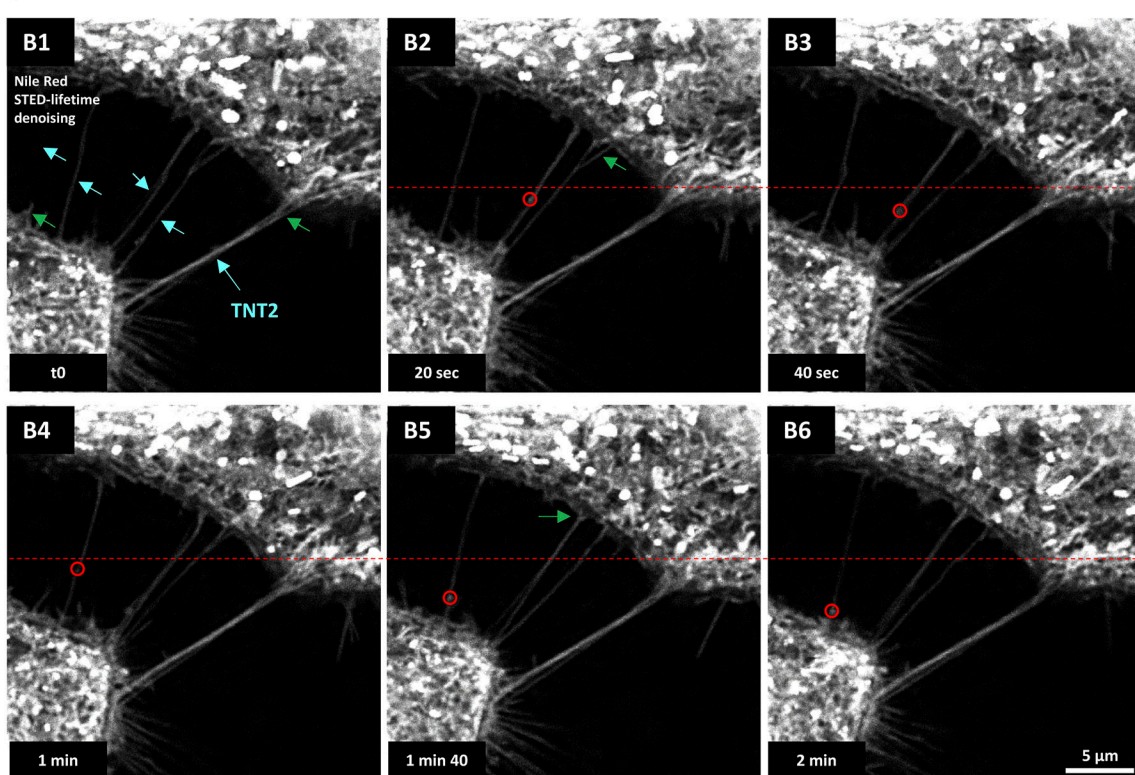

**Figure 5. Live-cell imaging of tunneling nanotubes (TNTs) between Nile Red–labeled H28 cells through fluorescence lifetime imaging microscopy–integrated stimulated emission depletion nanoscopy and lifetime denoising.**
Nile Red signal (ex 600 nm, 3% WLL laser power—HyD-X 615–670 nm) was depleted with 2% of a 775-nm pulsed laser and processed with lifetime denoising. **(A)** Time-lapse (10 min, 1 image/20 s, maximum intensity projection of a three-image z-stack) revealing mobile Nile Red–positive puncta (red circles) within TNT1 (A1–A3). **(B)** Time-lapse (2 min, 1 image/20 s, single frame) highlighting multiple TNT2 (blue arrows) with branched attachments (green arrows) and mobile Nile Red–positive elements (red circles). Reference red dotted lines facilitated the observation of positions during migration.

Act. A single 775-nm STED depletion laser (2%) was used. The SiR700-Act signal in HyD-X was denoised (STED lifetime denoising) and used for the final merged image, whereas the MTR signal in HyD-S was used for lifetime imaging (Fig 7 B6). The resulting merged image was therefore a combination of lifetime denoising and lifetime imaging. Interestingly, MTR-short and MTR-high lifetime values were associated with tubular- and round-shaped mitochondria, respectively (Fig 7 B4 and B6).

Collectively, these data indicated that TNT1 imaging through STED nanoscopy could be performed by activating two detectors and by combining lifetime denoising with lifetime unmixing or lifetime imaging when H28 cells were double-labeled.

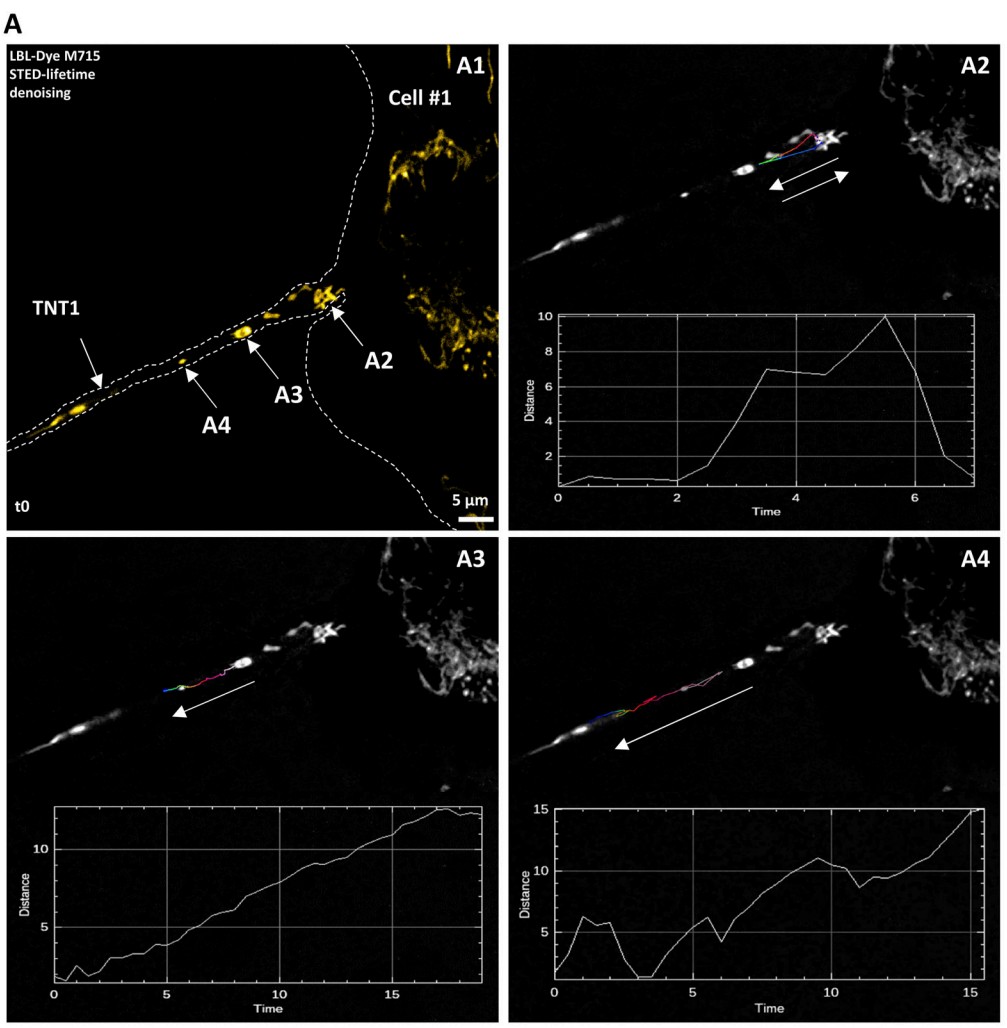

## FLIM-integrated STED nanoscopy - 1 dye - living H28 cells

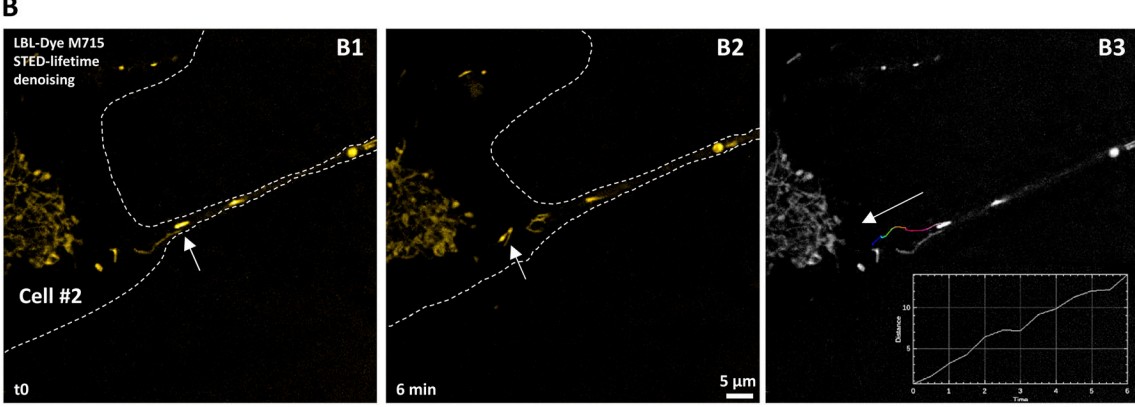

**Figure 6. Dynamics of LBL-Dye M715–labeled mitochondria within TNT1 connecting live H28 cells through stimulated emission depletion imaging and lifetime denoising.**
LBL-Dye M715 signal (ex 690 nm, 3% WLL laser power—HyD-X 705–740 nm) was depleted with 2% of a 775-nm pulsed laser and processed with lifetime denoising. **(A)** Time-lapse (20 min, 1 image/30 s) showing the trajectory of three mitochondria (A2, A3, and A4) within TNT1 emerging from the Cell #1 body (arrows, A1–A4). **(A, B)** Following (A), time-lapse (20 min, 1 image/30 s) showing the trajectory of 1 mitochondrion within TNT1 reaching the Cell #2 body. Graphs represented the distance (μm) from the origin to the end time points (min) of tracked objects. Trajectories (Fire LUT) were obtained using Fiji software, for example, the plugin Manual Tracking. A white dotted line delimitated the cell body and associated TNT1.

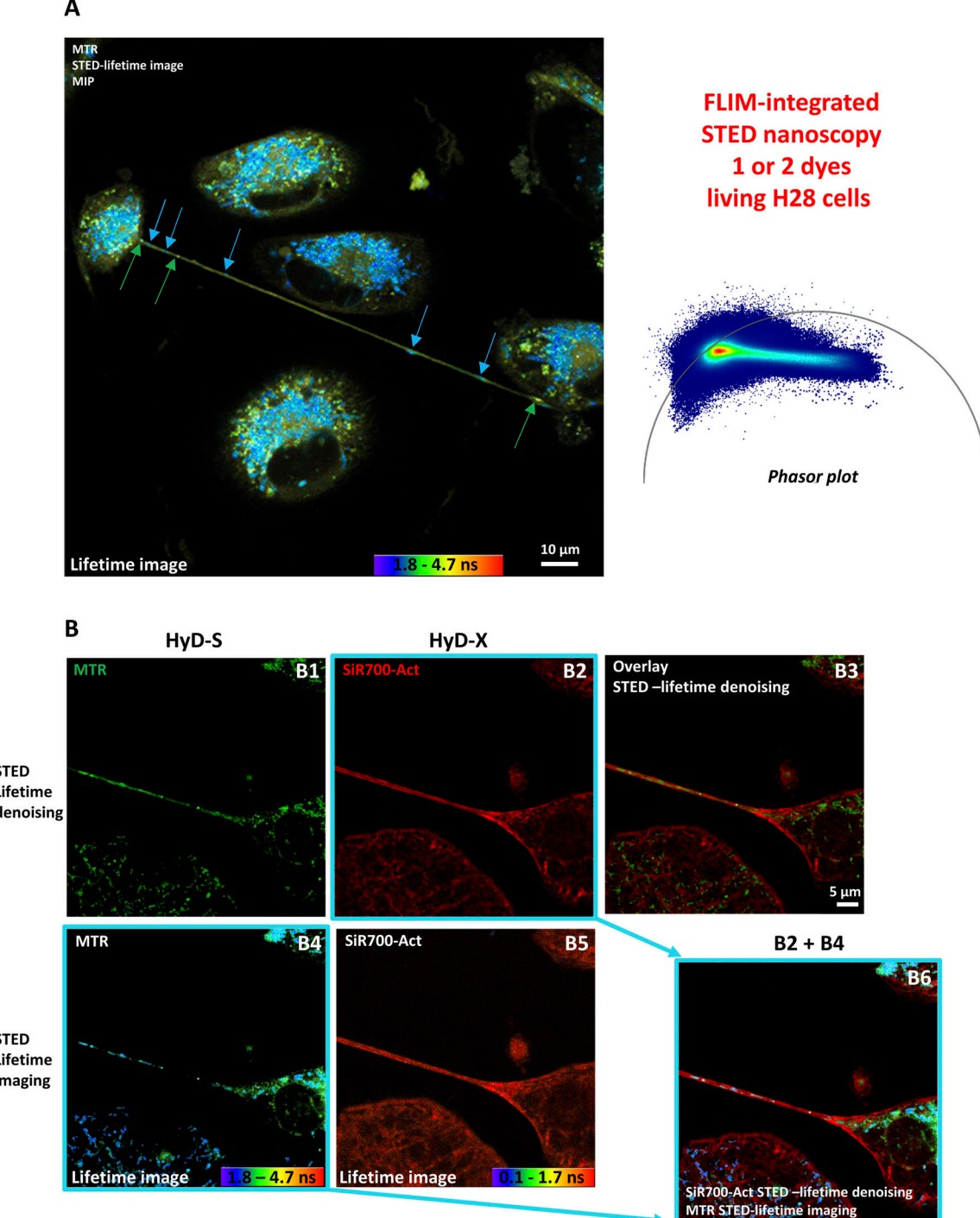

**Figure 7. Live-cell imaging of TNT1 between double-labeled H28 cells through fluorescence lifetime imaging microscopy–integrated stimulated emission depletion (STED) nanoscopy, lifetime dye unmixing, and lifetime denoising.**
**(A)** STED lifetime imaging of MitoTracker Red (MTR)–labeled H28 cells (ex 590, 1% WLL laser power—HyD-X 605–630 nm) and resulting maximum intensity projection of a five-image z-stack, 0.23 μm/step. The distribution of lifetime values was represented through a phasor plot with short values (1.3–1.8 ns) and high values (3.6–3.7 ns) positioned on right and left parts of the universal circle, respectively. **(B)** Simultaneous STED imaging of MTR (ex 590, 1% WLL laser power—HyD-S 605–630 nm) and SiR700-Act (ex 698, 6% WLL laser power—HyD-X 710–740 nm) signals depleted with 2% of a 775-nm laser. STED images were processed via lifetime denoising (B1-B2-B3) or lifetime imaging (B4-B5). Multi-lifetime components of the MTR signal were merged with lifetime denoising STED image of actin (B6).

# Discussion

Since its first description in 2004 (Rustom et al, 2004), cell biologists have taken a growing interest in the cell-to-cell communication via TNTs. Consequently, TNTs have been shown to connect many types of cell lines and primary cells (Cordero Cervantes & Zurzolo, 2021). Complementary biological models including co-culture and 3D culture have also revealed the diversity of TNTs including the existence of homo- and hetero-TNTs between similar or different cell types, respectively (Dubois et al, 2018; Dagar et al, 2021; Saha et al, 2022). Structural and functional characterizations of TNTs have been performed through complementary electron and advanced light microscopies (Dubois et al, 2021). Although requiring long time for sample preparation, TEM and SEM approaches have the advantage of access to the ultrastructure of TNTs with nanometric spatial resolution (Cordero Cervantes & Zurzolo, 2021). In contrast, membranes, organelles, lipids, and cytoskeleton proteins could be rapidly labeled with fluorescent markers in fixed cells and observed through confocal microscopy, SIM, or STED nanoscopy (Dubois et al, 2021). Considering the cellular, physical, and dynamic properties of TNTs, advanced live-cell imaging is challenging but necessary to achieve a full understanding of their mechanism of formation and transfer while preserving the sample from light exposure. Consequently, we implemented sophisticated strategies using red/near-infrared dyes and combined FLIM, confocal, and STED approaches to reveal TNTs in living mesothelial H28 cells.

## The quatuor "sample, probe, instrument, and image processing" as a guideline to propose innovative tools

As previously described (Galas et al, 2018), we considered the quatuor "sample," "probe," "instrument," and "image processing" as a successful guideline to obtain appropriate resolutions for fluorescence live-cell imaging. In this study, a pleural mesothelial H28 cell line was used as a cancer cell model to develop innovative advanced light tools to study the TNT features. Among various human epithelial bronchial or mesothelial cell lines, H28 cells do not express RASSF1A (Ras association domain family isoform) because the *RASSF1A* promoter gene is methylated (Dubois et al, 2018). The loss of RASSF1A expression, a key regulator of the cytoskeleton, is involved in TNT formation in bronchial and pleural cells by controlling the RhoB guanine nucleotide exchange factor and GEF-H1 activity (Dubois et al, 2018). Consequently, cultured mesothelial H28 cells spontaneously exhibit TNT1 and TNT2 (Dubois et al, 2018, 2020).

In order to preserve the viability of H28 cells during imaging via a reduced excitation/depletion laser power, we focused on red and near-infrared dyes and paid careful attention to their photophysical and STED properties, concentrations, and incubation time to optimize the right dye balance during multiplexing experiments (Bénard et al, 2021). For actin detection in living H28 cells, SPY650-FastAct and SiR700-Act were employed for single- and multi-labeling experiments. SPY650-FastAct is described as a non-toxic F-actin dye with the ability to label very fast actin dynamics. SiR700-actin is based on the silicon rhodamine (SiR) fluorophore analogue SiR700 and the actin-binding natural product jasplakinolide. To

increase loading, the SiR700 staining procedure of living H28 cells included a co-incubation with verapamil that is a broad-spectrum efflux pump inhibitor. WGA has been one of the most common markers for TNT membranes since early studies (Pasquier et al, 2012). In our study, we used WGA for membrane labeling at a concentration of 1 µg/ml during 15 min because similar conditions did not induce any significant increase in TNT formation (Pedicini et al, 2018). Interestingly, higher concentrations of WGA have been reported to increase the number of TNTs between endothelial cells ($EC_{50}$ = 8.17 µg/ml) (Pedicini et al, 2018). In addition, the used concentration of Alexa Fluor 633–conjugated WGA, the duration of incubation, and the imaging time after incubation may influence the role of WGA staining, from plasma membrane delimitation to intracellular labeling after membrane recycling. Nile Red stains typically the lipids including triglycerides, cholesterol and cholesteryl esters, and some phospholipids and is compatible with STED imaging (Bénard et al, 2021). Both cell-permeant lipophilic fluorescent LBL-Dye M715 and the derivative of X-rosamine MTR CMXRos were used for the labeling of mitochondria within TNTs and H28 cell bodies. In our experimental conditions, LBL-Dye M715 displayed a single-lifetime component, whereas MTR CMXRos, which fluoresces upon oxidation, showed a gradient of lifetimes.

The instrumental performances, namely, the sensitivity, the signal transmittance, and the stability, play a central role in fluorescence live-cell imaging with a particular focus on the key elements including objectives, laser sources, and detectors. In this work, we used a water immersion 86x objective (NA = 1.2) that displayed high capacities of transmission in the red/near-infrared range and high lateral resolution through STED nanoscopy leading consequently to low irradiance values (Bénard et al, 2021). The pulsed 775-nm laser depleted the red/near-infrared dyes in STED nanoscopy and induced less irradiance of living cells compared with the other continuous depletion lasers (Bénard et al, 2021). The new generation of high-speed photon counting detectors, known as hybrid detectors (HyD), brought striking opportunities for high signal-to-noise ratio and fast detection of fluorescence signals within the red/near-infrared range (Bénard et al, 2021). In this live-cell imaging study, we used preferentially HyD-X, which has a better sensitivity to the red spectrum and adapted temporal and physical characteristics for the fluorescence lifetime measurements. When two detectors were needed in double-labeling experiments through STED nanoscopy, a second detector HyD-S was selected because it displayed a rather good sensitivity all over the spectrum (Schweikhard et al, 2020; Bénard et al, 2021). The new generation of HyD detectors, combined with fast electronics, also offered new perspectives for multiplexing and lifetime acquisition in living cells including user-friendly solutions, and FLIM became consequently more accessible to cell biologists (Alvarez et al, 2019; Bitton et al, 2021; Gonzalez Pisfil et al, 2022). In order to bypass the spectral dye overlapping during multiplexing, the fluorescence lifetime separation was an alternative to the limited fluorescence spectral separation. Regardless of the excitation wavelength or light exposure duration, fluorescence lifetime is not affected by the photobleaching but may be influenced by factors such as viscosity, temperature, and pH (Berezin & Achilefu, 2010; Datta et al, 2020). Therefore, we proposed an agile methodology to prefer a fast simultaneous versus a sequential acquisition and to further operate

post-processing of the fluorescence lifetime collected data for STED denoising and lifetime unmixing via the phasor plot method. With the longest and shortest lifetimes located on the left and right parts of a universal circle, respectively, the phasor plot analysis provided a 2D graphical view of lifetime distributions (Digman & Gratton, 2012). The application of appropriate regions of interest within the universal circle enabled fine lifetime unmixing, and the phasor approach simplified consequently the FLIM image analysis.

Practically, it was necessary to first activate the FLIM module to integrate the fluorescence lifetime data within confocal or STED image metadata allowing lifetime imaging and lifetime unmixing through FLIM-integrated confocal microscopy and STED nanoscopy. The main difference between FLIM-integrated imaging approaches was that the rule "one dye–one detector–one phasor plot universal circle" had to be followed for STED nanoscopy, whereas it was much more flexible for confocal microscopy. Therefore, the live-cell multiplexing coupled to dye unmixing (phasor plot) was facilitated (the number of dyes, limited light exposure) and faster through FLIM-integrated confocal microscopy activating one and eventually several detectors (Frei et al, 2022), whereas it was more limited in STED nanoscopy. Sufficient differences in the fluorescent dye lifetime values were limiting factors for the dye combining and the phasor plot analysis. The STED nanoscopy images could be improved via lifetime unmixing to remove the background noise and via a specific lifetime denoising to increase the signal-to-noise ratio. Lifetime imaging of dyes with mono- or multi-lifetime components could be performed with both confocal and STED imaging. However, the lifetime imaging with multiple dyes was more easily performed when each dye displayed a mono-lifetime component or when only one dye among several had multi-lifetime components. For dyes with multi-components, the distribution and the discrimination of lifetime values were revealed by a rainbow look-up table.

For robust lifetime separation, researchers should first consider sufficient photon counting. Then, dyes with a mono-exponential component and a compacted phasor plot profile lying along the universal circle would be more easily separated even with close lifetime values (please see Fig S3; ~0.4 ns between SiR700-Act and LBL-Dye M715). In contrast, dyes with a multi-exponential component lifetime and a broadly spread phasor plot profile within the universal circle will be separated with more difficulty from other dyes. Consequently, researchers should empirically (i) carefully characterize the phasor plot profile of each dye in single-labeled cells before multiplexing experiments and (ii) make an appropriate combination of dyes depending on mono/multi-component lifetime. Interestingly, a recent publication has reported simultaneous imaging of nine fluorescent proteins in a single acquisition using FLIM combined with interleaved excitation of three laser lines (Starling et al, 2023).

The circumvention of the fluorescence live-cell imaging constraints including the probe specificity, the phototoxicity, the spectral overlapping, and the appropriate spatial and temporal resolutions drove the right decision for the acquisition as a single image or as stacks including short, mid-, or long time-lapses. Using the FLIM-STED imaging approach, the duration imaging is first related to photophysical properties of dyes including their brightness, their photostability, and the efficiency of depletion laser

for the S1-to-S0 dye transition. Indeed, a dye with high brightness and very compatible with STED imaging will require less excitation/depletion laser power, leading consequently to higher photostability over time and longer imaging time. In addition, the lower the laser power will be, the better the living sample will be preserved from light exposure. As an example of a mono-labeled case, LBL-Dye M715, which displays very interesting photophysical characteristics, allows time-lapse experiments that can long for 20 min (Video 6). The applications of our agile methodology for imaging TNTs in mesothelial H28 cells were therefore proposed below.

## Combining FLIM-integrated confocal and STED imaging to reveal TNT1 and TNT2 in living H28 cells

As previously described (Dubois et al, 2018), mesothelial H28 cells exhibited both TNT1 and TNT2 that contained differentially cytoskeleton proteins. In fixed H28 cells, actin, tubulin, and vimentin were observed in TNT1, whereas TNT2 only displayed actin labeling, suggesting differential structural and functional features related to the cortical actin, the microtubules, and the intermediate filaments. In particular, these data suggested distinct cargoes and transfer mechanisms between TNT1 and TNT2. In living cells, actin labeling with SPY650-FastAct or SiR700-Act was also useful for the delimitation of TNTs and the localization of organelles in multi-labeling experiments. As in several cell lines (Dubois et al, 2021), TNTs were revealed in living H28 cells through WGA-membrane labeling, which stained not only plasma membrane but also vesicle-like structures, particularly in large bud-shaped protrusion during TNT1 formation. Interestingly, fluorescence lifetime imaging highlighted the different types/states of vesicle-like structures. Because the fluorescence lifetime is sensitive to pH (Datta et al, 2020), the data suggest that FLIM-integrated confocal microscopy could monitor the variations of intravesicular pH. Moreover, both TNT1 and TNT2 were labeled by Nile Red, which fluoresces in the presence of a broad spectrum of lipids (Boumelhem et al, 2022). Intriguingly, an unusual Nile Red structure with an accordion-like shape was observed in TNT1, whereas staining was linear in TNT2. Such an accordion-like appearance was previously reported in H28 and HBEC-3 cells for tubulin that was potentially associated with micronuclei or DNA trail (Dubois et al, 2020). Interestingly, TNT-connected mesothelioma cells were significantly enriched with lipid rafts (Thayanithy et al, 2014). Numerous Nile Red–positive punctiform organelles, possibly lipid droplets, endosomes, or lysosomes, were detected in H28 cell bodies; however, only a few of them were found in TNT1 and TNT2. Previous studies also demonstrated that lipid droplets were considered as a cargo of TNTs (Astanina et al, 2015). The average speed of puncta in TNT2 was six times more than in TNT1. Lipid droplets were known to be able to associate with most other cellular organelles through membrane contact sites (Olzmann & Carvalho, 2019). In TNT1 of H28 cells, the rich cytoplasmic contents including actin, tubulin, vimentin, WGA-positive vesicle-like structures, and mitochondria may facilitate interaction of Nile Red–positive puncta with organelles reducing consequently their speed of movement. In contrast, TNT2 only contained actin and were virtually devoid of mitochondria and WGA-positive vesicular structures leading to possibly faster

movements. In addition, cholesterol/sphingomyelin liposomes can be transported via TNTs between glioblastoma U87-MG cells and were potentially useful as a drug delivery route for cancer therapy (Formicola et al, 2019). Mitochondria are highly dynamic organelles undergoing fission and fusion to maintain their shape, distribution, and size (Friedman & Nunnari, 2014; Tilokani et al, 2018). In addition, TNTs are increasingly recognized as a main intercellular route for unidirectional and bidirectional mitochondrial exchanges (Shanmughapriya et al, 2020; Chakraborty et al, 2023; Dong et al, 2023). In our experimental conditions, living mitochondria were revealed by LBL-Dye M715 and MTR, and only localized in TNT1. It was also reported that thinner TNTs were responsible for the short-distance transport of mitochondria, whereas thicker microtubule-containing TNTs were necessary for long-distance transport (MacAskill & Kittler, 2010; Wang & Gerdes, 2015; Zampieri et al, 2021). Within TNT1, both MTR-labeled tubular- and round-shaped mitochondria were detected and associated with short and high fluorescence lifetimes, respectively, suggesting differential metabolic states. In addition, the mitochondrial exchange was mainly unidirectional between H28 cells. Future multiplexing experiments with additional probes, such as SPY650-tubulin, SiR652-tubulin, endoplasmic reticulum, and lysosome and peroxisome probes, could also be considered.

In this study, several innovative tools have been developed to reveal TNTs in living cancer cells by setting up new protocols for single- and multi-labeling experiments using red/near-infrared dyes, and by taking advantages of the new instrumental performances. As a key element of the strategy, the exploitation of fluorescence lifetime data clearly improved functional considerations, dye separation, and signal-to-noise ratio. Therefore, we could move a step forward from fixed cell imaging to several minutes of time-lapse imaging while preserving the sample via the limitation of laser power. Combining sophisticated FLIM, confocal, and STED techniques has also provided novel insights about the differential structural and functional characteristics between TNT1 and TNT2 and their respective role in cell-to-cell communication. These tools will be very helpful for deciphering TNT formation steps and transfer mechanisms between cells including the impacts of stressful conditions and pharmacological treatments on mitochondrial dynamics and metabolic states. Future progress in TNT imaging will also be related to the development of new red/near-infrared probes performed by chemists. In addition to TNT, broader cell biology topics could also benefit from these advanced light imaging approaches that preserve living samples.

# Materials and Methods

## Cell culture

Purchased from the ATCC, pleural mesothelial H28 cell lines were cultured in RPMI-1640 (Thermo Fisher Scientific, Illkirch-Graffenstaden) medium supplemented with 2 mM of L-glutamine, 10% heat-inactivated FBS, 100 U/ml penicillin, and 100 µg/ml streptomycin (Thermo Fisher Scientific), as previously described (Bénard et al, 2021). Cultured cells were incubated at 37°C in a humidified atmosphere with 5% $CO_2$. For imaging experiments, H28

cells were plated at a density of $0.4 \times 10^4$ cells per $cm^2$ on 35-mm glass-bottom microwell dishes (MatTek Corporation).

## Cell labeling

For tubulin staining, H28 cells were fixed with 4% PFA for 5 min. After 30-min exposure to 1% BSA in PBS, cells were incubated at 4°C overnight with a monoclonal mouse antibody directed against tubulin (1:1,000; Merck) and a polyclonal rabbit antibody directed against vimentin (1:400, Cell Signaling; Merck) in PBS supplemented with 1% BSA and 0.3% Triton X-100. Cells were rinsed in PBS three times for 5 min and incubated for 2 h at RT with Alexa Fluor 594 donkey anti-rabbit (Thermo Fisher Scientific) and Atto 647N–conjugated goat anti-mouse (Merck) antibodies both diluted 1:400 in PBS supplemented with 1% BSA and 0.3% Triton X-100. After rinsing with PBS, F-actin was detected by Alexa Fluor 488–conjugated phalloidin (165 nM; Invitrogen) after 30-min incubation in PBS containing 1% BSA. Then, cells were rinsed in PBS three times for 5 min and coverslips were mounted with PBS/glycerol (50:50).

For live-cell imaging, H28 cells were labeled at 37°C/5% $CO_2$ with single or multiple dyes including (i) SPY620-DNA (0.5 µM) and/or SiR700-actin (0.5 µM) for 1 h (Tebu) together with verapamil (10 µM) or SPY650-FastAct (0.5 µM) for 1 h (Tebu), (ii) MitoTracker Red (0.2 µM) for 30 min (Thermo Fisher Scientific), (iii) LBL-Dye M715 (2 µg/ml) for 30 min (Proimaging), (iv) Nile Red (1 µM) for 10 min (Merck), and (v) Alexa Fluor 633–conjugated WGA (1 µg/ml) for 15 min (Thermo Fisher Scientific).

## Advanced confocal light scanning microscope

An inverted confocal laser scanning microscope (STELLARIS 8, Leica Microsystems) equipped with a white light laser (440–790 nm), four hybrid detectors (HyD type S, X, and R), an 86x objective (NA = 1.20, water immersion, WD = 300 µm) for living cell acquisition, a 100x objective (NA = 1.40, oil immersion, WD = 130 µm) for fixed cells, and a conventional scanner (400 Hz, 1,024 × 1,024) and Airy 1 pinhole was used. A fully fast integrated FLIM module, the so-called "FAst Lifetime CONtrast" (FALCON, Leica Microsystems), was used for confocal and STED acquisitions. A STED module (Leica Microsystems) with 592- and 775-nm pulsed depletion lasers was used to perform nanoscopy, the 775-nm depletion laser being necessary in live-cell conditions. For image acquisition, the appropriate zoom factor and pixel size were set in coherence with samples. For living cell experiments, a full bold line Okolab chamber (Ottaviano) installed on the inverted microscope stand was used to keep the temperature at 37°C and $CO_2$ at 5% during image acquisition.

## FLIM-integrated confocal imaging of TNTs in living H28 cells

A FLIM module was activated for the determination of fluorescence lifetime with 1-3 line repetition. FLIM-integrated confocal imaging of mono- or multi-labeled living H28 cells was performed with WLL lines (618, 638, or 650 nm; 2–5% AOTF) for dye excitation and the activation of a HyD-X in a photon counting mode for the detection of fluorescence collected through an 86x (NA = 1.2) objective. A

conventional scanner (400 Hz, 1,024 × 1,024) and Airy 1 pinhole were used. A FLIM module was activated for the determination of fluorescence lifetime with 1-3 line repetition to reach a minimum of 50,000 photons per image.

### FLIM-integrated STED imaging of TNTs in fixed and living H28 cells

A STED module (Leica Microsystems) installed on a STELLARIS 8 confocal base integrating two depletion lasers was used in this study. A 592-nm continuous depletion laser was dedicated to fluorochromes whose emission is in the range from 470 to 550 nm. A 775-nm pulsed depletion laser was dedicated to fluorochromes with an emission between 580 nm and 750 nm.

For fixed cells, the images were acquired in a sequential mode. As a first sequence, Alexa Fluor 594–conjugated vimentin and Atto 647N–conjugated tubulin were excited at 600 nm (WLL, 7% laser power) and 647 nm (WLL, 7%), respectively, and both depleted with 20% of a 775-nm STED laser. In a photon counting mode, HyD-S and HyD-X were used to detect fluorescence emission from 595 to 630 nm for Alexa Fluor 594 and from 660 to 710 nm for Atto 647N, respectively. In a second step, Alexa Fluor 488–conjugated phalloidin was excited at 488 nm (WLL, 5% laser power) and depleted with 40% of a high-energy 592-nm STED depletion laser. In a photon counting mode, a HyD-S was used to detect fluorescence emission from 500 to 540 nm. Acquisitions were performed through a 1,024 × 1,024 image format, a scan speed at 100 Hz, and a 100x objective, in a sequential stack-by-stack scan mode with a two-line accumulation, and a maximum intensity projection was obtained. 30% STED-3D (Z) was performed to improve the axial resolution, and resulting images were visualized thanks to 3D viewer software (Leica Microsystems).

For living cells, the STED images were acquired in a simultaneous mode. Fluorescence was collected through an 86x (NA = 1.2) objective. A conventional scanner (400 Hz, 1,024 × 1,024), Airy 1 pinhole, and a 775-nm depletion laser (2%) were used. STED imaging of mono-labeled H28 cells (e.g., Nile Red, LBL-Dye M715, or MTR) was performed with a single WLL line (600 or 690 nm, 1–6% AOTF) for dye excitation, and an activation of a HyD-X for fluorescence detection. For double-labeled living H28 cells, two excitation wavelengths and two HyD detectors were activated, for example, MTR (ex 590, 1% WLL laser power—HyD-S 605–630 nm) and SiR700-Act (ex 698, 6% WLL laser power—HyD-X 710–740 nm).

### Image analysis

Tracking and fluorescence intensity were measured using "Manual Tracking" plugin of Fiji software (Schindelin et al, 2012); trajectories were drawn in Fire LUT; and 3D reconstructions were performed using the "3D viewer" plugin of Fiji software and LAS X (Leica Microsystems).

### Statistical analysis

All values are expressed as a median (for asymmetric distribution) or means ± SEM. Statistical analysis was performed using GraphPad Prism 4 software (GraphPad Software Inc.) and a one-way ANOVA with a Tukey–Kramer multiple comparisons test.

## Supplementary Information

## Acknowledgements

This work was supported by the University of Rouen Normandy, Inserm, IRIB, Région Normandie (RIN plate-forme « 7D microscopy », RIN émergent COMVOI), the European Regional Development Fund (ERDF « 7D Microscopy »), the GIS IBiSA, and France BioImaging.

### Author Contributions

M Bénard: conceptualization, resources, formal analysis, validation, investigation, visualization, methodology, and writing—original draft, review, and editing.
C Chamot: conceptualization, data curation, formal analysis, validation, investigation, methodology, and writing—original draft, review, and editing.
D Schapman: conceptualization, formal analysis, validation, investigation, methodology, and writing—review and editing.
A Debonne: resources and writing—original draft, review, and editing.
A Lebon: data curation, formal analysis, validation, and writing—review and editing.
F Dubois: resources and writing—review and editing.
G Levallet: resources and writing—review and editing.
H Komuro: conceptualization and writing—review and editing.
L Galas: conceptualization, supervision, funding acquisition, validation, visualization, methodology, project administration, and writing—original draft, review, and editing.

### Conflict of Interest Statement

The authors declare that they have no conflict of interest.

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
