## [Reviewer comments · Life Science Alliance]

Life Science Alliance

Sophisticated fast FLIM, confocal and STED combining for live-cell imaging of tunneling nanotubes

Magalie Benard, Christophe Chamot, Damien Schapman, Aurélien Debonne, Alexis Lebon, Fatéméh Dubois, Guenaelle Levallet, Hitoshi Komuro, and Ludovic Galas

DOI: <https://doi.org/10.26508/lsa.202302398>

Corresponding author(s): Ludovic Galas, University of Rouen and Magalie Benard, Rouen University

Review Timeline:

Submission Date:	2023-09-27
Editorial Decision:	2023-12-04
Revision Received:	2024-02-09
Editorial Decision:	2024-02-29
Revision Received:	2024-04-01
Editorial Decision:	2024-04-02
Revision Received:	2024-04-09
Accepted:	2024-04-10

Transaction Report:

December 4, 2023

Re: Life Science Alliance manuscript #LSA-2023-02398-T

Dr. Ludovic Galas
University of Rouen
Cellular and Molecular Neuroendocrinology
UFR Sciences et Techniques, 25, rue Lucien Tesnière
Rouen 76000
France

Dear Dr. Galas,

Thank you for submitting your manuscript entitled "Sophisticated FLIM, confocal and STED combining for live-cell imaging of tunneling nanotubes" to Life Science Alliance. The manuscript was assessed by expert reviewers, whose comments are appended to this letter. We invite you to submit a revised manuscript addressing the Reviewer comments.

Thank you for this interesting contribution to Life Science Alliance. We are looking forward to receiving your revised manuscript.

Sincerely,

B. MANUSCRIPT ORGANIZATION AND FORMATTING:

Reviewer #1 (Comments to the Authors (Required)):

The paper titled "Sophisticated FLIM, confocal and STED combining for live-cell imaging of tunneling nanotubes" by Bénard et al. presents an approach that combines red/near-infrared dyes with FLIM, confocal, and STED microscopy techniques to investigate the characteristics of tunneling nanotubes (TNTs) in living H28 cells. While the authors provide extensive details on the applications of these microscopy tools for studying TNTs, it is still somewhat unclear what new insights can be gained from this method. In both the Introduction and Discussion sections, the authors acknowledge previous studies that have utilized light microscopy, including 3D-SIM and FLIM-FRET, to explore the dynamic properties of TNTs in cancer cells. Therefore, it is important to clarify which aspects of the demonstration in this manuscript represent newly discovered findings compared to previously reported results. To improve clarity regarding the unique discoveries made possible by this approach, the manuscript should be reorganized accordingly.

Here are some suggestions for revisions:

- a) The study utilizes techniques such as the combination of FLIM and STED, phasor analysis, and lifetime dye unmixing, which have previously been employed in the study of cellular structures and microenvironments. To highlight the advantages of these techniques in studying TNTs, it is crucial for the authors to provide a more comprehensive explanation of the novel insights that can be gained through this methodology specifically in the context of TNT research.
- b) In the 4th and 5th parts of the Results section, the authors demonstrate FLIM-STED microscopy with single and double infrared dyes, respectively. It is unclear why these two parts are separated. It appears that the time-lapse imaging demonstrated in mono-labeled cells could be integrated into the double-labeling experiments, providing a more cohesive presentation of the results.
- c) Lines 178-179 (Figure 3A): the authors state that "Alexa Fluor 633-WGA lifetime imaging was rather monocolour within TNT2, indicating a more homogeneous microenvironment." However, it is difficult to discern from the figure itself that the lifetime components in TNT1 are more heterogeneous than in TNT2. Further clarification or additional supporting evidence may be necessary to substantiate this claim.
- d) Lines 188-189 (Figure 3B): the authors mention that "the application of the phasor plot tool allowed the identification of two spots of lifetime values at 1.2 and 2.8 ns for LBL-Dye M715 and SPY650-FastAct, respectively." It would be beneficial to provide information on the minimum difference in lifetime values that can be reliably distinguished by the phasor plot tool. This knowledge would assist researchers in selecting suitable dyes for accurate lifetime dye separation.
- e) It would be valuable to include details regarding the longest imaging time achieved using the FLIM-STED imaging approach for both mono- and double-labeled cases, respectively. Additionally, it would be helpful to discuss the factors that may impose limitations on the imaging duration, providing insights into the constraints of the technique.
- f) Please note the presence of typos such as "environnement" (lines 175, 179).

Reviewer #2 (Comments to the Authors (Required)):

In this manuscript, the authors showcased STED, FLIM, and confocal images regarding tunneling nanotubes (TNTs). Each acquisition setting was carefully determined, so they provided supportive perspectives for future research using their configurations. However, I think this manuscript requires additional data to satisfy the aims and scope of this journal due to a lack of biological/methodological novelty and quantitative analysis. The following comments point out issues to be improved about the provided data.

Firstly, the authors optimized the experimental settings to conduct STED-FLIM imaging, which is indeed sophisticated. However, it had already been investigated well in their previous report (PMID: 34681761). Therefore, in terms of methodology, STED-FLIM is no longer claimed as a novel imaging system without additional installations or uncovering biological questions that can not be resolved with the previous techniques. If the authors claim that the signal-to-noise ratio (manuscript line 45) or spatial resolution was improved by the established configurations in this manuscript, they should demonstrate them with quantitative analysis such as intensity profiles. Moreover, descriptions about denoising in this manuscript are insufficient. The author should clearly describe further detail in method. Is the "STED-lifetime denoising" identical to Leica Tau-STED imaging?

Next, they applied their configuration to explore biological discoveries regarding TNTs in live-cells. Although they exemplified its

versatility using some fluorescent dyes, each observation needs to be supported by additional biological validations and contains some concerns.

Regarding statements about WGA imaging (manuscript line 176-179), the authors claimed the variance of lifetime is different between TNT1 and TNT2. However, to me, such a difference is unclear, and it seems just because of fewer photons in TNT2. To clarify these statements, the authors should quantify this observation with statistical analysis. A phasor plot of AF633-WGA in this manuscript (Fig. S2B) is not consistent with that in their previous report (Fig. 6 in PMID: 34681761) even though both were acquired with the same imaging configuration (FLIM). Does it imply any scientific curiosities? According to the reference (PMID: 29765077), WGA itself seems to induce TNT formations, implying WGA changes physiological conditions of TNTs. It can be better to mention it.

As for Nile red staining, the authors previously mentioned that this fluorescent dye has two states of lifetime in their previous report (PMID: 34681761). Nevertheless, they didn't mention it in this manuscript. The authors should provide an explicit explanation about "Nile red-positive elements" (whether they are droplets or not) with lifetime and STED-lifetime imaging with a rainbow look-up table as in other figures. The authors described a velocity of Nile red positive elements in line 237. However, the total number of objects and statistical analysis are missing. If the average speed of Nile red positive element transport shows a significant difference between TNT1 and TNT2, it can be helpful to discuss a functional difference between them.

Finally, the authors moved onto mitochondrial imaging in TNTs. As in Nile red measurements, the authors should provide total number and variance of speed of the LBL-Dye M715-positive mitochondria. The authors should provide a phasor plot of MitoTracker Red (MTR) and evaluate a correlation between lifetime and membrane potential (A control will be an addition of an uncoupling reagent such as FCCP). This can be helpful to argue about mitochondrial metabolic state in TNTs and lifetime as they mentioned in the manuscript line 277 and 441-443.

Point-by-point responses to reviewers:

Reviewer #1 (Comments to the Authors (Required)):

The paper titled "Sophisticated FLIM, confocal and STED combining for live-cell imaging of tunneling nanotubes" by Bénard et al. presents an approach that combines red/near-infrared dyes with FLIM, confocal, and STED microscopy techniques to investigate the characteristics of tunneling nanotubes (TNTs) in living H28 cells. While the authors provide extensive details on the applications of these microscopy tools for studying TNTs, it is still somewhat unclear what new insights can be gained from this method. In both the Introduction and Discussion sections, the authors acknowledge previous studies that have utilized light microscopy, including 3D-SIM and FLIM-FRET, to explore the dynamic properties of TNTs in cancer cells. Therefore, it is important to clarify which aspects of the demonstration in this manuscript represent newly discovered findings compared to previously reported results. To improve clarity regarding the unique discoveries made possible by this approach, the manuscript should be reorganized accordingly.

Here are some suggestions for revisions:

- a) **The study utilizes techniques such as the combination of FLIM and STED, phasor analysis, and lifetime dye unmixing, which have previously been employed in the study of cellular structures and microenvironments. To highlight the advantages of these techniques in studying TNTs, it is crucial for the authors to provide a more comprehensive explanation of the novel insights that can be gained through this methodology specifically in the context of TNT research.**

As recommended by the reviewer, we have highlighted the advantages of the "Sophisticated fast FLIM, confocal and STED combining for live-cell imaging of tunneling nanotubes".

We would like to thank the reviewer for his general comment. In order to clarify newly discovered findings, please first consider that 3D-SIM experiments have been only performed on fixed, not living, urothelial cells (Resnik et al., 2018). This 3D-SIM approach was reported in the Introduction paragraph dedicated to fixed samples studies. Secondly, Hanna et al. used a wide-field imaging device for their FRET-based Rho GTPase biosensors experiments with consequently limitations in spatial resolution. Thirdly, Wang et al. studied protein transmission between living cells through a slow TCSPC FLIM-FRET home-made device (PMID: 33996210, Figure 3). However, they did not show any time-lapse experiments. The field-programmable gate array (FGPA) in our FLIM module enables "fast FLIM" by directly measuring differences in arrival times between detection and excitation pulses and consequently time-lapses. Consequently, our technological strategy overcomes the limitations of the previous reports.

We have replaced all over the text "FLIM" by "fast FLIM" and we have modified the abstract, introduction result and conclusion sections as follows:

Abstract: Page 2

Thanks to a fast FLIM module integrated to confocal microscopy and STED nanoscopy, we applied lifetime imaging, lifetime dye unmixing, and lifetime denoising techniques to perform multiplexing experiments and time-lapses of tens of minutes revealing therefore structural and functional characteristics of living TNTs that were preserved from light exposure. In these conditions, vesicle-like structures, tubular and rounded-shape mitochondria were identified within living TNT1.

Introduction: Pages 3 and 4

In fixed cancer urothelial cells, 3D-SIM experiments revealed the presence of microtubules (MT) and intermediate filaments (IF) within TNTs, with MTs potentially forming a helical wrapping around IFs (Resnik et al., 2018).

Wide-field FRET-based Rho GTPase biosensors revealed distinct activation pattern of Cdc42 and Rac1 at the base and within TNTs in macrophage cell line (Hanna et al., 2017). In addition, home-made two-photon excitation fluorescence lifetime imaging microscopy (TCSPC-FLIM)-FRET was used to detect material transport via TNTs in ovarian cancer cells (Wang et al., 2021).

In addition, missing reference Wang et al., 2021b (PMID: 33996210) was added to the reference list.

Result section: Page 6

Since offering multiple options in confocal and STED acquisition modes, the integrated fast FLIM module was definitively a central and revolutionary tool in our strategy and Figure 2 was proposed as a guide for readers to perform labeling and long-lasting time-lapse experiments for studying structural and functional aspects of living TNTs including organelle trafficking.

Conclusion: Page 19

Overcoming SIM and TCSPC FLIM limitations, we therefore could move a step forward from imaging of fixed cells to several minutes-time-lapses experiments while preserving sample via the reduction of laser power. Combining sophisticated fast FLIM, confocal and STED techniques has also provided novel insights about the differential structural and functional characteristics between TNT1 and TNT2 and their respective role in cell-to-cell communication. These tools will be very helpful for deciphering TNT formation steps and transfer mechanisms between cells including the impacts of stressful conditions and pharmacological treatments on mitochondria dynamics and metabolic states. Future progress in TNT imaging will also be related to the development of new red/near-infrared probes performed by chemists. In addition to TNT, broader cell biology topics could also benefit from these advanced-light imaging approaches that preserve living samples.

Additional information were also added in Figure 2 to emphasize the input of these techniques to TNT research including preservation from light exposure during time-lapses of tens of minutes. Legend to Figure 2 was also consequently modified.

- b) In the 4th and 5th parts of the Results section, the authors demonstrate FLIM-STED microscopy with single and double infrared dyes, respectively. It is unclear why these two parts are separated. It appears that the time-lapse imaging demonstrated in mono-labeled cells could be integrated into the double-labeling experiments, providing a more cohesive presentation of the results.**

We would like to thank the reviewer for his comment that make the manuscript easier for readers. The 4th and the 5th parts of the Results section in the original version were merged in a single section so called: "Fast FLIM-integrated STED nanoscopy and lifetime denoising revealed TNTs dynamics in living H28 cells labeled with 1 or 2 red/infrared dyes".

- c) Lines 178-179 (Figure 3A): the authors state that "Alexa Fluor 633-WGA lifetime imaging was rather monocolored within TNT2, indicating a more homogeneous microenvironment." However, it is difficult to discern from the figure itself that the lifetime components in TNT1 are more heterogeneous than in TNT2. Further clarification or additional supporting evidence may be necessary to substantiate this claim.**

We would like to thank the reviewer for his comments helping us to improve the manuscript.

Former Figure 3A2 has been replaced by a zoomed image illustrating particularly the dispersion of Alexa Fluor 633-WGA lifetime values along TNT1 and within large bud-shaped of TNT1. Additional information were also provided in supplementary Figure S2. First, the phasor plot profile of the entire lifetime image of Figure 3B2 was indeed broadly spread out compare to the phasor plot profile of the entire lifetime image of Figure 3B3 (Figure S2 B1 and C1). Such phasor plot profiles were also observed in region of interest delimiting specifically TNT1 and TNT2 (Figure S2 B2 and C2). Please also note that specific TNT1 and TNT2 phasor plot profiles were obtained with very close number of photons *i.e.* 50 192 for TNT1 ROI and 50 619 for TNT2 ROI. Altogether, the dispersion profiles of lifetime values in TNT1 ROI suggested the presence of WGA-positive vesicle-like structures that could not be detected in TNT2.

To clarify our analysis, the text in the Result section has been modified page 7 and now reads:

Interestingly, the dispersion of Alexa Fluor 633-WGA lifetimes was broader along TNT1 and within the large bud-shaped (Fig 3A2) compare to that observed in TNT2 (Fig 3A3). TNT1 and TNT2 lifetime phasor plot profiles were different (Fig S2B and C). Indeed, TNT1 ROI displayed higher Alexa Fluor 633-WGA lifetime values compare to TNT2 ROI

(Fig S2B2 and C2). Altogether, these data suggest the presence of WGA-positive vesicle-like structures in TNT1 but not in TNT2.

Original Figure S2 was split in Figure S2 and in a new Figure S3. Legend to Figure 3, Figure S2 and Figure S3 were also consequently modified and *in fine* 6 supplementary figures are presented.

d) Lines 188-189 (Figure 3B): the authors mention that "the application of the phasor plot tool allowed the identification of two spots of lifetime values at 1.2 and 2.8 ns for LBL-Dye M715 and SPY650-FastAct, respectively." It would be beneficial to provide information on the minimum difference in lifetime values that can be reliably distinguished by the phasor plot tool. This knowledge would assist researchers in selecting suitable dyes for accurate lifetime dye separation.

We would like to thank the reviewer for his interesting comment. Researchers have to consider several parameters for dye selection and consequently accurate lifetime dye separation. First, sufficient photon counting is of main importance for robust dye separation. Second, mono- or multi-exponential component of lifetime could be observed in a cellular or subcellular environment. Dyes with mono-exponential component and compacted phasor plot profile lying along the universal circle would be more easily separated even with closed lifetime values (please see Figure S2 C and D; ~0.4 ns between SiR700-Act and LBL-Dye M715). In contrast, dyes with multi-exponential component lifetime and broadly spread phasor plot profile within the universal circle will be separated more difficulty from other dyes. Consequently, researchers should empirically i) carefully characterize the phasor plot profile of each dye in single-labeled cells before multiplexing experiments and ii) make appropriate combination of dye depending of mono/multi-component lifetime. Therefore, it is rather difficult to provide strict information on the minimum difference in lifetime values for appropriate separation. Interestingly, a recent publication has reported simultaneous imaging of nine fluorescent proteins in a single acquisition using FLIM combined with interleaved excitation of three laser lines (Starling et al, 2023).

We added in the discussion section the following section page 16:

For robust lifetime separation, researchers should first consider sufficient photon counting. Then, dyes with mono-exponential component and compacted phasor plot profile lying along the universal circle would be more easily separated even with closed lifetime values (please see Figure S3; ~0.4 ns between SiR700-Act and LBL-Dye M715). In contrast, dyes with multi-exponential component lifetime and broadly spread phasor plot profile within the universal circle will be separated more difficulty from other dyes. Consequently, researchers should empirically i) carefully characterize the phasor plot profile of each dye in single-labeled cells before multiplexing experiments and ii) make appropriate combination of dye depending of mono/multi-component lifetime. Interestingly, a recent publication has reported simultaneous imaging of nine fluorescent proteins in a single acquisition using FLIM combined with interleaved excitation of three laser lines (Starling et al, 2023).

In addition, **Starling et al., 2023** was added to the reference list.

e) It would be valuable to include details regarding the longest imaging time achieved using the FLIM-STED imaging approach for both mono- and double-labeled cases, respectively. Additionally, it would be helpful to discuss the factors that may impose limitations on the imaging duration, providing insights into the constraints of the technique.

To satisfy the reviewer request, additional sentences have been inserted within the Discussion section page 17

Using the fast FLIM-STED imaging approach, the duration imaging is first related to photophysical properties of dyes including their brightness, their photostability and the efficiency of depletion laser for S1 to S0 dye transition. Indeed, a dye with high brightness and very compatible to STED imaging will require less excitation/depletion laser power leading consequently to higher photostability over time and longer imaging time. In addition, the lower the laser power will be, better the living sample will preserved from light exposure. As an example, labeling H28 cells with LBL-Dye M715 which displays very interesting photophysical characteristics allows 20 min-time-lapse experiments (video 6).

f) Please note the presence of typos such as "environnement" (lines 175, 179).

- We would like to apologize for this mistake. Consequently, environnement (lines 175 and 179) was replaced by **environment**
- Similarly, revolutionary was replaced by **revolutionary** (page 6)

Point-by-point responses to reviewers:

Reviewer #2 (Comments to the Authors (Required)):

In this manuscript, the authors showcased STED, FLIM, and confocal images regarding tunneling nanotubes (TNTs). Each acquisition setting was carefully determined, so they provided supportive perspectives for future research using their configurations. However, I think this manuscript requires additional data to satisfy the aims and scope of this journal due to a lack of biological/methodological novelty and quantitative analysis. The following comments point out issues to be improved about the provided data.

Firstly, the authors optimized the experimental settings to conduct STED-FLIM imaging, which is indeed sophisticated. However, it had already been investigated well in their previous report (PMID: 34681761). Therefore, in terms of methodology, STED-FLIM is no longer claimed as a novel imaging system without additional installations or uncovering biological questions that can not be resolved with the previous techniques. If the authors claim that the signal-to-noise ratio (manuscript line 45) or spatial resolution was improved by the established configurations in this manuscript, they should demonstrate them with quantitative analysis such as intensity profiles. Moreover, descriptions about denoising in this manuscript are insufficient. The author should clearly describe further detail in method. Is the "STED-lifetime denoising" identical to Leica Tau-STED imaging?

As mentioned in the introduction, this study is dedicated to live-cell imaging of tunneling nanotube (TNT) which is very challenging due to the thinness, fragility and dynamic of such structures. The approach was never performed previously for TNT and as far as we know, we are the only one in the world performing such imaging for living TNTs. The previous paper (PMID: 34681761) was definitively a proof of concept for living sample studies and here we go further by improving combination of fluorescent probes, reducing laser power, performing long time-lapses to focus our study on TNTs.

In the previous paper the improvement of signal-to-noise ratio (SNR) and spatial resolution were already demonstrated in particular with fluorescent beads through quantitative analysis (PMID: 34681761, including supplementary files). Since the purpose of our current study was to preserve TNTs by reducing as much as possible light exposure, it is unfortunately not appropriated to perform successive acquisitions through confocal, fast FLIM and STED on living samples, that are necessary to demonstrate increase in SNR and resolution. Furthermore, successive acquisitions were even impossible when organelles within TNTs moved or displayed shape-changing steps or both. In this study, we focused on TNT research with the best imaging compromise between spatial and temporal resolution, laser power and sample preservation.

Sometimes confusing, different approaches so called Tau approaches were described by the Leica Microsystems company. Without any fast FLIM module, you can actually performed TauSense for confocal microscopy which is in fact very constrained and not opened to post-processing (no fitting, no phasor plot analysis). In our previous paper, we described this approach as a “coarse τ separation” because we did not want to be tightly linked to Leica marketing. Similarly ‘coarse’ TauSTED (TauSens without the FALCON module) can be performed but we consider that it was not fully appropriate to control lifetime data. Thanks to the fast FLIM integrated module (FALCON), we prefer to perform “fine τ separation”, a more robust approach for both confocal and STED imaging also named FLIM, phasor and TauSTED by Leica Microsystems. In our paper submitted to Life Alliance Science, we emphasized the fine τ separation approaches and named fast FLIM-integrated confocal (or fast FLIM-confocal) and fast FLIM-integrated STED imaging (or fast FLIM-STED) instead of FLIM, phasor and Tau-STED. Lifetime data obtained from fast FLIM-integrated STED imaging allow image post-processing that we called “Lifetime denoising”. In our study, “Lifetime denoising” included i) removal of uncorrelated STED process photons (background subtraction, weighting of lifetime) and ii) signal smoothing through a wavelet filter.

Consequently, we would like to thank the reviewer for this comment making the manuscript easier for readers. To clearly describe “denoising”, additional sentence was added in the Result section, page 6:

In our study, “Lifetime denoising” included *i*) removal of uncorrelated STED process photons (background subtraction, weighting of lifetime) and *ii*) signal smoothing through a wavelet filter.

Next, they applied their configuration to explore biological discoveries regarding TNTs in live-cells. Although they exemplified its versatility using some fluorescent dyes, each observation needs to be supported by additional biological validations and contains some concerns.

The article has been submitted has a “Methods” article in order to propose an experimental guide for readers to study living TNTs. As also discussed later, these approaches will open the possibility to explore in future studies the processes of TNT formation and cell-to-cell transfer in physiological and stressful conditions. Consequently, we will consider the relevant reviewer comment for future experiments.

Regarding statements about WGA imaging (manuscript line 176-179), the authors claimed the variance of lifetime is different between TNT1 and TNT2. However, to me, such a difference is unclear, and it seems just because of fewer photons in TNT2. To clarify these statements, the authors should quantify this observation with statistical analysis.

We would like to thank the reviewer for his comments helping us to improve the manuscript.

Former Figure 3A2 has been replaced by a zoomed image illustrating particularly the dispersion of Alexa Fluor 633-WGA lifetime values along TNT1 and within large bud-

shaped of TNT1. Additional information were also provided in supplementary Figure S2. First, the phasor plot profile of the entire lifetime image of Figure 3B2 was indeed broadly spread out compare to the phasor plat profile of the entire lifetime image of Figure 3B3. Such phasor plot profiles were also observed in region of interest delimiting specifically TNT1 and TNT2. Please also note that specific TNT1 and TNT2 phasor plot profiles were obtained with very close number of photons *i.e.* 50 192 for TNT1 ROI and 50 619 for TNT2 ROI. Altogether, the dispersion profiles of lifetime values in TNT1 ROI suggested the presence of WGA-positive vesicle-like structures that could not be detected in TNT2.

To clarify our analysis, the text in the Result section has been modified page 7 and now reads:

Interestingly, the dispersion of Alexa Fluor 633-WGA lifetimes was broader along TNT1 and within the large bud-shaped (Fig 3A2) compare to that observed in TNT2 (Fig 3A3). TNT1 and TNT2 lifetime phasor plot profiles were different (Fig S2B and C). Indeed, TNT1 ROI displayed higher Alexa Fluor 633-WGA lifetime values compare to TNT2 ROI (Fig S2B2 and C2). Altogether, these data suggest the presence of WGA-positive vesicle-like structures in TNT1 but not in TNT2.

In addition, the following sentence was added in the Materials and Methods section, page 22: "Fast FLIM module was activated for the determination of fluorescence lifetime with 1-3 line repetition to reach a minimum of 50 000 photons per image."

A phasor plot of AF633-WGA in this manuscript (Fig. S2B) is not consistent with that in their previous report (Fig. 6 in PMID: 34681761) even though both were acquired with the same imaging configuration (FLIM). Does it imply any scientific curiosities?

We would like to thank the reviewer for this relevant comment. The phasor plot of AF633-WGA in Fig. S2B was obtained from a single zoomed (x2) image that focuses on TNTs. In contrast, AF633-WGA phasor plot from Fig.6 in PMID: 34681761 resulted from a zoomless time-lapse (5 min, 70 images). Consequently, the lifetime distribution was more spread out due to larger field of view and compilation of several images obtained over time. As an information, please find below the phasor plot of AF633-WGA of Fig. S2B from full time-lapse experiment (5 min, 20 images) illustrating our demonstration.

According to the reference (PMID: 29765077), WGA itself seems to induce TNT formations, implying WGA changes physiological conditions of TNTs. It can be better to mention it.

We would like to thank the reviewer for the important comment. In our study, we used WGA for membrane labeling at a concentration of 1 µg/ml during 15 min. Similar conditions in PMID: 29765077- Figure 1 did not induce any significant increase in TNT formation between endothelial cells, a cell model that does not express spontaneously TNT. In contrast, our biological model namely H28 cells are spontaneously and highly connected through TNTs (Dubois et al, 2020). To satisfy the reviewer request, an additional sentence has been added in the discussion section page 14 as follows:

Interestingly, WGA has been reported to increase the number of TNT between endothelial cells ($EC_{50} = 8.17 \mu\text{g/ml}$) (Pedicini et al, 2018).

In addition, PMID: 29765077 was added to the reference list.

As for Nile red staining, the authors previously mentioned that this fluorescent dye has two states of lifetime in their previous report (PMID: 34681761). Nevertheless, they didn't mention it in this manuscript. The authors should provide an explicit explanation about "Nile red-positive elements" (whether they are droplets or not) with lifetime and STED-lifetime imaging with a rainbow look-up table as in other figures.

We did not use the same Nile Red supplier in (PMID: 34681761) (<https://www.enzolifesciences.com/ENZ-52551/nile-red-ultra-pure/>) and in this study (<https://www.sigmaaldrich.com/FR/fr/product/sigma/n3013?icid=sharepdp-clipboard-copy-productdetailpage>).

Indeed, the one from Enzo was prepared in DMSO and the one from Sigma in Methanol. Surprisingly, we could observed TNT labeling with Sigma-Nile Red while not with the Enzo one. We had discussions with chemists that could not unfortunately bring any explanation for different labeling with similar molecule. In addition, the Sigma-Nile Red phasor plot with a monoexponential lifetime (3.37 ns) for membrane and lipid droplet elements, is totally different from the phasor plot obtained in PMID: 34681761. Please find below STED lifetime imaging for TNT1 and TNT2 connecting H28 cells that illustrates monoexponential lifetime of Nile Red. Round-shape Nile-positive organelles are definitively lipid droplets. Identification of other Nile Red-positive signals that were detected along TNT1 and TNT2 required further investigation.

The authors described a velocity of Nile red positive elements in line 237. However, the total number of objects and statistical analysis are missing. If the average speed of Nile red positive element transport shows a significant difference between TNT1 and TNT2, it can be helpful to discuss a functional difference between them.

We would like to thank the reviewer for this comment helping to improve the manuscript. The average speed of Nile Red positive elements in TNT1 and TNT2 were $0.76 \pm 0.24 \mu\text{m}/\text{min}$ ($n=5$) and $4.86 \pm 0.6 \mu\text{m}/\text{min}$ ($n=5$) respectively.

Nile Red positive elements are likely to be lipid droplets known to be able to associate with most other cellular organelles through membrane contact sites (Olzmann et al., 2019). In TNT1 of H28 cells, the rich cytoplasmic content including actin, tubulin, vimentin, WGA-positive vesicle-like structures and mitochondria may facilitate interaction of lipid droplets with organelles reducing consequently their speed of movement. In contrast, TNT2 only contained actin and were virtually devoid of mitochondria and WGA-positive vesicular structures leading to possibly faster movements.

Additional sentences have been added in the Result section page 10 and now reads:

The average speed of Nile Red positive lipid droplets in TNT1 and TNT2 were $0.76 \pm 0.24 \mu\text{m}/\text{min}$ ($n=5$) and $4.86 \pm 0.6 \mu\text{m}/\text{min}$ ($n=5$) respectively.

Additional sentences have been added in the Discussion section page 18 and now reads:

The average speed of lipid droplets in TNT2 was 6 times more than in TNT1. Lipid droplets were known to be able to associate with most other cellular organelles through membrane contact sites (Olzmann et al., 2019). In TNT1 of H28 cells, the rich cytoplasmic content including actin, tubulin, vimentin, WGA-positive vesicle-like structures and mitochondria may facilitate interaction of lipid droplets with organelles reducing consequently their speed of movement. In contrast, TNT2 only contained actin and were virtually devoid of mitochondria and WGA-positive vesicular structures leading to possibly faster movements.

In addition, Olzmann et al., 2019 was added to the reference list.

Finally, the authors moved onto mitochondrial imaging in TNTs. As in Nile red measurements, the authors should provide total number and variance of speed of the LBL-Dye M715-positive mitochondria. The authors should provide a phasor plot of MitoTracker Red (MTR) and evaluate a correlation between lifetime and membrane potential (A control will be an addition of an uncoupling reagent such as FCCP). This can be helpful to argue about mitochondrial metabolic state in TNTs and lifetime as they mentioned in the manuscript line 277 and 441-443.

We would like to thank the reviewer for this relevant comment helping to improve the manuscript. As described initially in the results section, mitochondria displayed very different rate profiles. Due to such dispersion, we provided the median of rate profiles (1.435; n=24), associated with minimum (0.8 $\mu\text{m}/\text{min}$) and maximum (8.14 $\mu\text{m}/\text{min}$) values as well as 1st and 3rd quartile values (0.935 and 4.525 respectively).

Additional sentences have been added in the Result section page 9 and now reads:

The median speed of mitochondria within TNT1 was 1.435 (n=24) (min = 0.8 $\mu\text{m}/\text{min}$; max = 8.14 $\mu\text{m}/\text{min}$; n=24) while 1st and 3rd quartile values were 0.935 and 4.525 respectively.

As request by the reviewer, a phasor plot of MitoTracker Red was added in Figure 7A. Legend to Figure 7 was also consequently modified.

The mechanism of mitochondria transfer through TNT1 is of main importance to emphasize in particular pathophysiological considerations. In this technological and methodological paper, we wanted to demonstrate that live-cell imaging of mitochondria within TNT1 could be actually performed. Impacts of stressful conditions and pharmacological treatments on mitochondria dynamics and metabolic states will be considered in a dedicated future work.

Consequently, additional sentences have been added in the text page 19 and now reads:

Combining sophisticated fast FLIM, confocal and STED techniques has also provided novel insights about the differential structural and functional characteristics between TNT1 and TNT2 and their respective role in cell-to-cell communication. These tools will be very helpful for deciphering TNT formation steps and transfer mechanisms between cells including the impacts of stressful conditions and pharmacological treatments on mitochondria dynamics and metabolic states. Future progress in TNT imaging will also be related to the development of new red/near-infrared probes performed by chemists. In addition to TNT, broader cell biology topics could also benefit from these advanced-light imaging approaches that preserve living samples.

February 29, 2024

Re: Life Science Alliance manuscript #LSA-2023-02398-TR

Dr. Ludovic Galas
University of Rouen
HeRacLeS - PRIMACEN
UFR Sciences et Techniques, 25, rue Lucien Tesnière
Rouen 76000
France

Dear Dr. Galas,

Thank you for submitting your revised manuscript entitled "Sophisticated fast FLIM, confocal and STED combining for live-cell imaging of tunneling nanotubes" to Life Science Alliance. The manuscript has been seen by the original reviewers whose comments are appended below, and some important issues remain.

Our general policy is that papers are considered through only one revision cycle; however, we are open to one additional short round of revision. Please note that I will expect to make a final decision without additional reviewer input upon re-submission. The concerns expressed about the perceived novelty of the method is not an editorial concern, however the other comments should be addressed.

Please submit the final revision within one month, along with a letter that includes a point by point response to the remaining reviewer comments.

To upload the revised version of your manuscript, please log in to your account: <https://lsa.msubmit.net/cgi-bin/main.plex>
You will be guided to complete the submission of your revised manuscript and to fill in all necessary information.

B. MANUSCRIPT ORGANIZATION AND FORMATTING:

Sincerely,

Reviewer #1 (Comments to the Authors (Required)):

The concerns I raised previously have been adequately addressed by the authors. I do not have any additional comments regarding the manuscript at this time.

Reviewer #2 (Comments to the Authors (Required)):

The inclusion of additional information provided by the authors enhanced the clarity of the manuscript and bridged some gaps from their previous report. However, they did not provide the most crucial points raised in my earlier comments (sufficient novelty and quantification), resulting in my conclusion being unchanged, unfortunately.

While the authors claimed this manuscript as a method article, I could not find substantive modifications from existing methods in this manuscript. Their appeal on novelty is an application of their imaging configuration to a new biological subject, TNTs, indicating that this article is not methodologically new. Regrettably, "the first attempt" does not always ensure novelty without discoveries or substantive developments.

Regarding Fig. 3 and Fig. S2 (WGA imaging), the authors replaced to the zoomed images, but unfortunately, I do not see significant improvement. This further confused me about the presence of "bud (or vesicle)-shape" in TNT1. The authors should encircle a vesicle-like structure with optimal lines (white dot line etc.).

In terms of the lifetime variance of AF633-WGA between TNT1 and TNT2, phasor plots (Fig. S2B2 and C2) are likely helpful for the author's claim rather than micrographs. However, because quantification and statistical analysis were not provided, I cannot evaluate whether this observation is reproducible and to what extent variance (between samples) exists.

In author's comment, I could not get "the phasor plot profile of the entire lifetime image of Figure 3B2 was indeed broadly spread out compare to the phasor plot profile of the entire lifetime image of Figure 3B3". Figure 3B2 and 3B3 seem like the data of SPY650-FastAct, but not WGA.

Even though they explained as in "Please also note that specific TNT1 and TNT2 phasor plot profiles were obtained with very close number of photons i.e. 50 192 for TNT1 ROI and 50 619 for TNT2 ROI", they amended the manuscript as in "Fast FLIM module was activated for the determination of fluorescence lifetime with 1-3 line repetition to reach a minimum of 50 000 photons per image". Phasor plots (Fig. S2B2 and C2) seem to demonstrate fewer photons in the TNT2 ROI, please clarify it, and mention that it did not mean 50 000 photons per pixel (which seems excessively high for imaging).

Regarding the author's explanation about a discrepancy of AF633-WGA's lifetime distribution in a phasor plot between Fig. S2B (this manuscript) and Fig. 6 (PMID: 34681761), it could convince me about a difference in broadness, but it does not explain the reason why the average lifetime (the center of the population) is shifted.

Regarding my previous comment starting from "According to the reference (PMID: 29765077),...", the authors provided a satisfactory response in the point-by-point document. But they should also describe the same statements into the revised manuscript by summarizing their comments "In our study, we used WGA for membrane labeling at a concentration of 1 µg/ml during 15 min. Similar conditions in PMID: 29765077- Figure 1 did not induce any significant increase in TNT formation...". This statement can justify their WGA staining in optimal condition.

Regarding Nile Red -positive elements (Fig. 5), as far as I assessed them in the provided images and videos, unfortunately I could not conclude them as lipid droplets. The author's explanation "the Sigma-Nile Red phasor plot with a monoexponential lifetime (3.37 ns) for membrane and lipid droplet elements, ..." explicates that Sigma-Nile Red cannot distinguish membranes and lipid droplets. Given that Nile Red changes its fluorescent spectrum according to the microenvironment related to lipids (neutral/polar), its monoexponential lifetime leaves a suspicious concern about its characteristics. Considering the presence of obvious puncta in the cell body region (to me, they look like lipid droplets rather than Nile Red positive elements in TNTs, but it is disabled to distinguish them from endosomes and lysosomes), Nile Red positive elements in TNTs are likely presenting something different (including another possibility such as temporal swelling of the membrane reflecting transports of something big).

Point-by-point responses to reviewer #2:

Reviewer #2 (Comments to the Authors (Required)):

Regarding Fig. 3 and Fig. S2 (WGA imaging), the authors replaced to the zoomed images, but unfortunately, I do not see significant improvement. This further confused me about the presence of "bud (or vesicle)-shape" in TNT1. The authors should encircle a vesicle-like structure with optimal lines (white dot line etc.).

As previously described (PMID: **26094971**; PMID: **32351987**), TNT1 may exhibited in different cell line models a large trumpet-shaped origin, a clear cytosolic tunnel and different bud-shaped connections from closed-ended to open-ended tips. Here we showed, a dense accumulation of WGA-positive vesicle-like structures with dispersed and high Alexa Fluor-WGA lifetime values within the large bud-shaped of TNT1 (Figure 3A2, white arrow indicating bud-shaped protrusion of TNT1). The Figure 3A2 is a MIP of a 2.72 μm stack making consequently difficult to identify single vesicles. Therefore, and as request by the reviewer 2, additional figures have been inserted in Figure S2 to clarify this point. Indeed, blue line-encircled single vesicle-like structures were identified within 3 single intensity signal planes ($z_1 = 0$; $z_2 = 0.68 \mu\text{m}$; $z_3 = 1.36 \mu\text{m}$). In conclusion, Figures 3A2 and S2 and associated legends have been appropriately modified.

In terms of the lifetime variance of AF633-WGA between TNT1 and TNT2, phasor plots (Fig. S2B2 and C2) are likely helpful for the author's claim rather than micrographs. However, because quantification and statistical analysis were not provided, I cannot evaluate whether this observation is reproducible and to what extent variance (between samples) exists.

We are pleased to read that comparison of phasor plots from ROI TNT1 and TNT2 are helpful for the reviewer and for readers. We used phasor plot analyses to highlight dispersion and multi-exponential components of AF633-WGA in vesicle-like structures. Consequently, average lifetime would not be informative and appropriate. In addition, among cell lines studied so far by our group, feature of WGA-intensity signal in TNT1 and TNT2 was similar (PMID: **26094971**; PMID: **32351987**) i.e. plasma membrane and intracellular vesicle labeling in TNT1 depending on used concentration and incubation time. Here, we wanted to bring new information with fluorescence lifetime imaging to illustrate differences between TNT1 and TNT2.

In author's comment, I could not get "the phasor plot profile of the entire lifetime image of Figure 3B2 was indeed broadly spread out compare to the phasor plat profile of the entire lifetime image of Figure 3B3". Figure 3B2 and 3B3 seem like the data of SPY650-FastAct, but not WGA.

We would like to apologize for this mistake. In our answer, we should have written Figure 3A2 and Figure 3A3 instead of Figure 3B2 and 3B3. To make it clearer for reviewer 2:

- The phasor plot profile now in Figure S2B1 illustrates the entire lifetime image of Figure 3A2.
- The phasor plot profile now in Figure S2B3 illustrates the ROI TNT1 lifetime image of Figure S2B2.
- The phasor plot profile now in Figure S2C1 illustrates the entire lifetime image of Figure 3A3.
- The phasor plot profile now in Figure S2C3 illustrates the ROI TNT2 lifetime image of Figure S2C2.

In conclusion, Figure S2 and associated legend have been appropriately modified.

Even though they explained as in "Please also note that specific TNT1 and TNT2 phasor plot profiles were obtained with very close number of photons i.e. 50 192 for TNT1 ROI and 50 619 for TNT2 ROI", they amended the manuscript as in "Fast FLIM module was activated for the determination of fluorescence lifetime with 1-3 line repetition to reach a minimum of 50 000 photons per image". Phasor plots (Fig. S2B2 and C2) seem to demonstrate fewer photons in the TNT2 ROI, please clarify it, and mention that it did not mean 50 000 photons per pixel (which seems excessively high for imaging).

As previously described, the total number of photons in ROI TNT1 was 50 192. The total number of photons in TNT2 was 50 619. Both numbers were indicated now in Figure S2B2 and Figure S2C2.

Please note that in the Materials and Methods section it was mentioned: "Fast FLIM module was activated for the determination of fluorescence lifetime with 1-3 line repetition to reach a minimum of 50 000 photons per image." So, 50 000 photons/image and not by pixel.

In conclusion, Figure S2 and associated legend have been appropriately modified.

Regarding the author's explanation about a discrepancy of AF633-WGA's lifetime distribution in a phasor plot between Fig. S2B (this manuscript) and Fig. 6 (PMID: 34681761), it could convince me about a difference in broadness, but it does not explain the reason why the average lifetime (the center of the population) is shifted.

We are pleased that the reviewer was convinced by our previous explanations and we would like to thank the reviewer for his additional comment. As previously stated, the configuration of image acquisition was different from experiments in PMID: 34681761 (zoom, time-lapse, number of images). Used AF633-WGA batch was similar to our previous experiments but cell culture and bovine fetal serum batch were different suggesting limitations to strict comparison. Most importantly, vesicles-like structure with higher lifetime values could be detected in both works (this submitted paper and PMID: 34681761). In contrast, short fluorescence lifetimes could be mainly detected at

the membrane level in PMID: 34681761 (Figure A2, blue color in the FLIM scale 0-2.4 ns, 1.122 ns) while it was not the case in Figure 2 of the submitted work. Therefore, the center of the lifetime population in AF633-WGA labeled H28 cells could be shifted due to different experimental conditions and to different distribution of lifetime values within different images.

Regarding my previous comment starting from "According to the reference (PMID: 29765077),...", the authors provided a satisfactory response in the point-by-point document. But they should also describe the same statements into the revised manuscript by summarizing their comments "In our study, we used WGA for membrane labeling at a concentration of 1 µg/ml during 15 min. Similar conditions in PMID: 29765077- Figure 1 did not induce any significant increase in TNT formation...". This statement can justify their WGA staining in optimal condition.

We were pleased to provide a satisfactory response to the reviewer and we would like to thank him for his additional comment. Consequently, the statement in the revised manuscript now reads page 14: "In our study, we used WGA for membrane labeling at a concentration of 1 µg/ml during 15 min since similar conditions did not induce any significant increase in TNT formation (Pedicini et al., 2018). Interestingly, higher concentrations of WGA have been reported to increase the number of TNT between endothelial cells (EC50 = 8.17 µg/ml) (Pedicini et al., 2018).

Regarding Nile Red-positive elements (Fig. 5), as far as I assessed them in the provided images and videos, unfortunately I could not conclude them as lipid droplets. The author's explanation "the Sigma-Nile Red phasor plot with a monoexponential lifetime (3.37 ns) for membrane and lipid droplet elements, ..." explicates that Sigma-Nile Red cannot distinguish membranes and lipid droplets. Given that Nile Red changes its fluorescent spectrum according to the microenvironment related to lipids (neutral/polar), its monoexponential lifetime leaves a suspicious concern about its characteristics. Considering the presence of obvious puncta in the cell body region (to me, they look like lipid droplets rather than Nile Red positive elements in TNTs, but it is disabled to distinguish them from endosomes and lysosomes), Nile Red positive elements in TNTs are likely presenting something different (including another possibility such as temporal swelling of the membrane reflecting transports of something big).

We would like to thank the reviewer from his general comment. As described carefully in the original version (modified in revised version #1 due to reviewer comments), we have moderated the identification of Nile Red positive elements in the revised version including "Results" and "Discussion" sections and the Abstract to avoid any over interpretation.

Highlighted in yellow, modifications in revised version#1; highlighted in green, modifications in revised version #2.

Results section:

Using 775-nm STED-compatible Nile Red, which is usually described as a vital intracellular lipid marker, both TNT1 and TNT2 were revealed and time-lapses were performed over several minutes (Fig 5). Nile Red-positive signals were detected along the length of TNTs with wavelet/accordion-like appearances in TNT1 and straight aspects in TNT2 (Fig 5). Nile Red also labeled branched attachments of TNT2 that strengthened TNT2 to the cell body (green arrows, Fig 5 B). In the cytoplasm of cell bodies, Nile Red also labeled punctiform organelles that were sparsely found within TNT1 (Fig 5 A) and TNT2 (Fig 5 B). Time-lapse imaging clearly illustrated the movements of Nile Red-labeled puncta within TNT1 (red circles, 0.53 $\mu\text{m}/\text{min}$) and TNT2 (red circles, 4.6 $\mu\text{m}/\text{min}$) towards H28 cell bodies (Video 4 and Video 5, respectively). The average speed of Nile Red positive puncta in TNT1 and TNT2 were $0.76 \pm 0.24 \mu\text{m}/\text{min}$ (n=5) and $4.86 \pm 0.6 \mu\text{m}/\text{min}$ (n=5) respectively.

Discussion section:

Moreover, both TNT1 and TNT2 were labeled by Nile Red which fluoresces in the presence of a broad spectrum of lipids (Boumelhem et al., 2022). Intriguingly, an unusual Nile Red structure with an accordion-like shape was observed in TNT1, while staining was linear in TNT2. Such accordion-like appearance was previously reported in H28 and HBEC-3 cells for tubulin that was potentially associated with micronuclei or DNA trail (Dubois et al., 2020). Interestingly, TNT-connected mesothelioma cells were significantly enriched with lipid rafts (Thayanithy et al., 2014). Numerous Nile Red-positive punctiform organelles, possibly lipid droplets, endosomes or lysosomes, were detected in H28 cell bodies, however only a few of them were found in TNT1 and TNT2. Previous studies also demonstrated that lipid droplets were considered as a cargo of TNTs (Astaniina et al., 2015). The average speed of puncta in TNT2 was 6 times more than in TNT1. Lipid droplets were known to be able to associate with most other cellular organelles through membrane contact sites (Olzmann et al., 2019). In TNT1 of H28 cells, the rich cytoplasmic content including actin, tubulin, vimentin, WGA-positive vesicle-like structures and mitochondria may facilitate interaction of Nile Red-positive puncta with organelles reducing consequently their speed of movement. In contrast, TNT2 only contained actin and were virtually devoid of mitochondria and WGA-positive vesicular structures leading to possibly faster movements.

Abstract:

Transfer of Nile Red-positive puncta via both TNT1 and TNT2 were also detected between living H28 cells.

April 2, 2024

RE: Life Science Alliance Manuscript #LSA-2023-02398-TRR

Dr. Ludovic Galas
University of Rouen
HeRacLeS - PRIMACEN
UFR Sciences et Techniques, 25, rue Lucien Tesnière
Rouen 76000
France

Dear Dr. Galas,

Thank you for submitting your revised manuscript entitled "Sophisticated fast FLIM, confocal and STED combining for live-cell imaging of tunneling nanotubes". We would be happy to publish your paper in Life Science Alliance pending final revisions necessary to meet our formatting guidelines.

- please be sure that the authorship listing and order is correct
- please add ORCID ID for corresponding the secondary corresponding author -- they should have received instructions on how to do so
- please add the Twitter handle of your host institute/organization as well as your own or/and one of the authors in our system
- please incorporate any points from the Conclusion section into the Discussion, we only allow a Discussion section
- The Author Contributions selected for Fatéméh Dubois, Guenaëlle Levallet and Hitoshi Komur do not qualify them for authorship. Please update the contributions in the system and in the manuscript, or let us know if the authors should be removed.
- Figure S1 lacks section C in the figure itself, which is listed in the figure legend; Figure S5 lacks sections B4 and B6 in the figure caption
- please add callouts for Figures S1C; S2A-C and S5A-B to your main manuscript text

A. FINAL FILES:

B. MANUSCRIPT ORGANIZATION AND FORMATTING:

Sincerely,

April 10, 2024

RE: Life Science Alliance Manuscript #LSA-2023-02398-TRRR

Dr. Ludovic Galas
University of Rouen
HeRacLeS - PRIMACEN
UFR Sciences et Techniques, 25, rue Lucien Tesnière
Rouen 76000
France

Dear Dr. Galas,

Thank you for submitting your Methods entitled "Sophisticated fast FLIM, confocal and STED combining for live-cell imaging of tunneling nanotubes". It is a pleasure to let you know that your manuscript is now accepted for publication in Life Science Alliance. Congratulations on this interesting work.

DISTRIBUTION OF MATERIALS:

Again, congratulations on a very nice paper. I hope you found the review process to be constructive and are pleased with how the manuscript was handled editorially. We look forward to future exciting submissions from your lab.

Sincerely,
